# Stage-wise Conservative Linear Bandits

**Ahmadreza Moradipari, Christos Thrampoulidis, Mahnoosh Alizadeh**
Department of Electrical and Computer Enginnering
University of California, Santa Barbara
ahmadreza_moradipari@ucsb.edu

## Abstract

We study stage-wise conservative linear stochastic bandits: an instance of bandit optimization, which accounts for (unknown) "safety constraints" that appear in applications such as online advertising and medical trials. At each stage, the learner must choose actions that not only maximize cumulative reward across the entire time horizon, but further satisfy a linear baseline constraint that takes the form of a *lower bound* on the instantaneous reward. For this problem, we present two novel algorithms, *stage-wise conservative linear Thompson Sampling* (SCLTS) and *stage-wise conservative linear UCB* (SCLUCB), that respect the baseline constraints and enjoy probabilistic regret bounds of order $\mathcal{O}(\sqrt{T}\log^{3/2}T)$ and $\mathcal{O}(\sqrt{T}\log T)$, respectively. Notably, the proposed algorithms can be adjusted with only minor modifications to tackle different problem variations, such as, constraints with bandit-feedback, or an unknown sequence of baseline rewards. We discuss these and other improvements over the state-of-the art. For instance, compared to existing solutions, we show that SCLTS plays the (non-optimal) baseline action at most $\mathcal{O}(\log T)$ times (compared to $\mathcal{O}(\sqrt{T})$). Finally, we make connections to another studied form of "safety constraints" that takes the form of an *upper bound* on the instantaneous reward. While this incurs additional complexity to the learning process as the optimal action is not guaranteed to belong to the "safe set" at each round, we show that SCLUCB can properly adjust in this setting via a simple modification.

## 1 Introduction

With the growing range of applications of bandit algorithms for safety critical real-world systems, the demand for safe learning is receiving increasing attention Tucker et al. (2020). In this paper, we investigate the effect of stage-wise safety constraints on the linear stochastic bandit problem. Inspired by the earlier work of Kazerouni et al. (2017); Wu et al. (2016), the type of safety constraint we consider in this paper was first introduced by Khezeli and Bitar (2019). As with the classic linear stochastic bandit problem, the learner wishes to choose a sequence of actions $x_t$ that maximize the expected reward over the horizon. However, here the learner is also given a baseline policy that suggests an action with a guaranteed level of expected reward at each stage of the algorithm. This could be based on historical data, e.g., historical ad placement or medical treatment policies with known success rates. The safety constraint imposed on the learner requires her to ensure that the expected reward of her chosen action at every single round be no less than a predetermined fraction of the expected reward of the action suggested by baseline policy. An example that might benefit from the design of stage-wise conservative learning algorithms arises in recommender systems, where the recommender might wish to avoid recommendations that are extremely disliked by the users at any single round. Our proposed stage-wise conservative constraints ensure that at no round would the recommendation system cause severe dissatisfaction for the user, and the reward of action employed by the learning algorithm, if not better, should be close to that of baseline policy. Another example is

in clinical trials where the effects of different therapies on patients' health are initially unknown. We can consider the baseline policy to be treatments that have been historically employed, with known effectiveness. The proposed stage-wise conservative constraint guarantees that at each stage, the learning algorithm suggests an action (a therapy) that achieves the expected reward close to that of the baseline treatment, and as such, this experimentation does not cause harm to *any single patient's health*. To tackle this problem, Khezeli and Bitar (2019) proposed a greedy algorithm called SEGE. They use the decomposition of the regret first proposed in Kazerouni et al. (2017), and show an upper bound of order $\mathcal{O}(\sqrt{T})$ over the number of times that the learning algorithm plays the baseline actions, overall resulting in an expected regret of $\mathcal{O}(\sqrt{T}\log T)$. For this problem, we present two algorithms, SCLTS and SCLUCB, and we provide regret bounds of order $\mathcal{O}(\sqrt{T}\log^{3/2} T)$ and $\mathcal{O}(\sqrt{T}\log T)$, respectively. As it is explained in details in Section 3, we improve the result of Khezeli and Bitar (2019), i.e., we show our proposed algorithms play the (non-optimal) baseline actions at most $\mathcal{O}(\log T)$ times, while also relaxing a number of assumptions made in Khezeli and Bitar (2019). Moreover, we show that our proposed algorithms are adaptable with minor modifications to other safety-constrained variations of this problem. This includes the case where the constraint has a different unknown parameter than the reward function with bandit feedback (Section 3.1), as well as the setting where the reward of baseline action is unknown to the learner in advance (Section 4).

## 1.1 Conservative Stochastic Linear bandit (LB) Problem with Stage-wise Constraints

**Linear Bandit.** The learner is given a convex and compact set of actions $\mathcal{X} \subset \mathbb{R}^d$. At each round $t$, she chooses an action $x_t$ and observes a random reward

$$y_t = \langle x_t, \theta_\star \rangle + \xi_t, \tag{1}$$

where $\theta_\star \in \mathbb{R}^d$ is *unknown* but fixed reward parameter and $\xi_t$ is zero-mean additive noise. We let $r_t$ be the expected reward of action $x_t$ at round $t$, i.e., $r_t := \mathbb{E}[y_t] = \langle x_t, \theta_\star \rangle$.

**Baseline actions and stage-wise constraint.** We assume that the learner is given a baseline policy such that selecting the baseline action $x_{b_t}$ at round $t$, she would receive an expected reward $r_{b_t} := \langle x_{b_t}, \theta_\star \rangle$. We assume that the learner knows the expected reward of the actions chosen by the baseline policy. We further assume that the learner's action selection rule is subject to a stage-wise conservative constraint of the form[1]

$$r_t = \langle x_t, \theta_\star \rangle \geq (1-\alpha)r_{b_t}, \tag{2}$$

that needs to be satisfied at each round $t$. In particular, constraint (2) guarantees that at each round $t$, the expected reward of the action chosen by the learner stays above the predefined fraction $1-\alpha \in (0,1)$ of the baseline policy. The parameter $\alpha$, controlling the conservatism level of the learning process, is assumed known to the learner similar to Kazerouni et al. (2017); Wu et al. (2016). At each round $t$, an action is called *safe* if its expected reward is above the predetermined fraction of the baseline policy, i.e., $(1-\alpha)r_{b_t}$.

**Remark 1.1.** *It is reasonable to assume that the leaner has an accurate estimate of the expected reward of the actions chosen by baseline policy Kazerouni et al. (2017). However, in Section 4, we relax this assumption, and propose an algorithm to the case where the expected rewards of the actions chosen by baseline policy are unknown to the learner in advance.*

**Regret.** The *cumulative pseudo-regret* of the learner up to round $T$ is defined as $R(T) = \sum_{t=1}^{T} \langle x_\star, \theta_\star \rangle - \langle x_t, \theta_\star \rangle$, where $x_\star$ is the optimal safe action that maximizes the expected reward,

$$x_\star = \arg\max_{x \in \mathcal{X}} \langle x, \theta_\star \rangle. \tag{3}$$

The learner's objective is to minimize the pseudo-regret, while respecting the stage-wise conservative constraint in (2). For the rest of the paper, we use regret to refer to the pseudo-regret $R(T)$.

## 1.2 Previous work

**Multi-armed Bandits.** The multi-armed bandit (MAB) framework has been studied in sequential decision making problems under uncertainty. In particular, it captures the exploration-exploitation trade-off, where the learner needs to sequentially choose arms in order to maximize her reward over time while exploring to improve her estimate of the reward of each arm Bubeck and Eldan (2016). Two popular heuristics exist for MAB: Following the *optimism in face of uncertainty* (OFU) principle Auer et al. (2002); Li et al. (2017); Filippi et al. (2010), the so-called Upper Confidence Bound (UCB) based approaches choose the best feasible action- environment pair according to their current confidence regions on the unknown parameter, and Thompson Sampling (TS) Thompson (1933); Kaufmann et al. (2012); Russo and Van Roy (2016); Moradipari et al. (2018), which randomly samples the environment and plays the corresponding optimal action.

**Linear Stochastic Bandits.** There exists a rich literature on linear stochastic bandits. Two well-known efficient algorithms for LB are Linear UCB (LUCB) and Linear Thompson Sampling (LTS). For LUCB, Dani et al. (2008); Rusmevichientong and Tsitsiklis (2010); Abbasi-Yadkori et al. (2011) provided a regret guarantee of order $\mathcal{O}(\sqrt{T}\log T)$. For LTS, Agrawal and Goyal (2013); Abeille et al. (2017) provided a regret bound of order $\mathcal{O}(\sqrt{T}\log^{3/2} T)$ in a frequentist setting, i.e., when the unknown parameter $\theta_\star$ is a fixed parameter. We need to note that none of the aforementioned heuristics can be directly adopted in the conservative setting. However, note that the regret guarantee provided by our extensions of LUCB and LTS for the safe setting matches those stated for the original setting.

**Conservativeness and Safety.** The baseline model adopted in this paper was first proposed in Kazerouni et al. (2017); Wu et al. (2016) in the case of *cumulative constraints* on the reward. In Kazerouni et al. (2017); Wu et al. (2016), an action is considered feasible/safe at round $t$ as long as it keeps the cumulative reward up to round $t$ above a given fraction of a given baseline policy. This differs from our setting, which is focused on stage-wise constraints, where we want the expected reward of the *every single action* to exceed a given fraction of the baseline reward at each time $t$. This is a tighter constraint than that of Kazerouni et al. (2017); Wu et al. (2016). The setting considered in this paper was first studied in Khezeli and Bitar (2019), which proposed an algorithm called SEGE to guarantee the satisfaction of the safety constraint at each stage of the algorithm. While our paper is motivated by Khezeli and Bitar (2019), there are a few key differences: 1) We prove an upper bound of order $\mathcal{O}(\log T)$ for the number of times that the learning algorithm plays the conservative actions which is an order-wise improvement with respect to that of Khezeli and Bitar (2019), which shows an upper bound of order $\mathcal{O}(\sqrt{T})$; 2) In our setting, the action set is assumed to be a general convex and compact set in $\mathbb{R}^d$. However, in Khezeli and Bitar (2019), the proof relies on the action set being a specific ellipsoid; 3) In Section 4, we provide a regret guarantee for the learning algorithm for the case where the baseline reward is unknown. However, the results of Khezeli and Bitar (2019) have not been extended to this case; 4) In Section 3.1, we also modify our proposed algorithm and provide a regret guarantee for the case where the constraint has a different unknown parameter than the one in the reward function. However, this is not discussed in Khezeli and Bitar (2019). Another difference between the two works is on the type of performance guarantees. In Khezeli and Bitar (2019), the authors bound the *expected* regret. Towards this goal, they manage to quantify the effect of the risk level $\delta$ on the regret and constraint satisfaction. However, it appears that the analysis in Khezeli and Bitar (2019) is limited to ellipsoidal action sets. Instead, in this paper, we present a bound on the regret that holds with high (constant) probability (parameterized by $\delta$) over *all* $T$ rounds of the algorithm. This type of results is very common in the bandit literature, e.g. Abbasi-Yadkori et al. (2011); Dani et al. (2008), and in the emerging safe-bandit literature Kazerouni et al. (2017); Amani et al. (2019); Sui et al. (2018).

Another variant of safety w.r.t a baseline policy has also been studied in Mansour et al. (2015); Katariya et al. (2018) in the multi-armed bandits framework. Moreover, there has been an increasing attention on studying the effect of safety constraints in the Gaussian process (GP) optimization literature. For example, Sui et al. (2015, 2018) study the problem of *nonlinear* bandit optimization with nonlinear constraints using GPs (as non-parametric models). The algorithms in Sui et al. (2015, 2018) come with convergence guarantees but no regret bound. Moreover, Ostafew et al. (2016); Akametalu et al. (2014) study safety-constrained optimization using GPs in robotics applications. A large body of work has considered safety in the context of model-predictive control, see, e.g., Aswani et al. (2013); Koller et al. (2018) and references therein. Focusing specifically on linear stochastic

bandits, extension of UCB-type algorithms to provide safety guarantees with provable regret bounds was considered recently in Amani et al. (2019). This work considers the effect of a linear constraint of the form $x^\top B\theta_\star \leq C$, where $B$ and $C$ are respectively a known matrix and positive constant, and provides a problem dependent regret bound for a safety-constrained version of LUCB that depends on the location of the optimal action in the safe action set. Notice that this setting requires the linear function $x^\top B\theta_\star$ to remain below a threshold $C$, as opposed to our setting which considers a lower bound on the reward. We note that the algorithm and proof technique in Amani et al. (2019) does not extend to our setting and would only work for inequalities of the given form; however, we discuss how our algorithm can be modified to provide a regret bound of order $\mathcal{O}(\sqrt{T}\log T)$ for the setting of Amani et al. (2019) in Appendix H. A TS variant of this setting has been studied in Moradipari et al. (2020); Moradipari et al. (2019).

## 1.3 Model Assumptions

**Notation.** The weighted $\ell_2$-norm with respect to a positive semi-definite matrix $V$ is denoted by $\|x\|_V = \sqrt{x^\top V x}$. The minimum of two numbers $a, b$ is denoted $a \wedge b$. Let $\mathcal{F}_t = (\mathcal{F}_1, \sigma(x_1, \xi_1, \ldots, x_t, \xi_t))$ be the filtration ($\sigma$-algebra) that represents the information up to round $t$.

**Assumption 1.** *For all $t$, $\xi_t$ is conditionally zero-mean R-sub-Gaussian noise variables, i.e., $\mathbb{E}[\xi_t|\mathcal{F}_{t-1}] = 0$, and $\mathbb{E}[e^{\lambda\xi_t}|\mathcal{F}_{t-1}] \leq \exp(\frac{\lambda^2 R^2}{2}), \forall \lambda \in \mathbb{R}^d$.*

**Assumption 2.** *There exists a positive constant $S$ such that $\|\theta_\star\|_2 \leq S$.*

**Assumption 3.** *The action set $\mathcal{X}$ is a compact and convex subset of $\mathbb{R}^d$ that contains the unit ball. We assume that $\|x\|_2 \leq L, \forall x \in \mathcal{X}$. Also, we assume $\langle x, \theta_\star \rangle \leq 1, \forall x \in \mathcal{X}$.*

Let $\kappa_{b_t} = \langle x_\star, \theta_\star \rangle - r_{b_t}$ be the difference between expected reward of the optimal and baseline actions at round $t$. As in Kazerouni et al. (2017), we assume the following.

**Assumption 4.** *There exist $0 \leq \kappa_l \leq \kappa_h$ and $0 < r_l \leq r_h$ such that, at each round $t$*

$$\kappa_l \leq \kappa_{b_t} \leq \kappa_h \text{ and } r_l \leq r_{b_t} \leq r_h. \tag{4}$$

We note that since these parameters are associated with the baseline policy, it can be reasonably assumed that they can be estimated accurately from data. This is because we think of the baseline policy as "past strategy", implemented before bandit-optimization, thus producing large amount of data. The lower bound $0 < r_l \leq r_{b_t}$ on the baseline reward ensures a minimum level of performance at each round. $\kappa_h$ and $r_h$ could be at most 1, due to Assumption 3. For simplicity, we assume the lower bound $\kappa_l$ on the sub-optimality gap $\kappa_{b_t}$ is known. If not, we can always choose $\kappa_l = 0$ by optimality of $x_\star$.

## 2 Stage-wise Conservative Linear Thompson Sampling (SCLTS) Algorithm

In this section we propose a TS variant algorithm in a frequentist setting referred to as *Stage-wise Conservative Linear Thompson Sampling* (SCLTS) for the problem setting in Section 1.1. Our adoption of TS is due to its well-known computational efficiency over UCB-based algorithms, since action selection via the latter involves solving optimization problems with bilinear objective functions, whereas the former would lead to linear objectives. However, this choice does not fundamentally affect our approach. In fact, in Appendix G, we propose a Stage-wise Conservative Linear UCB (SCLUCB) algorithm, and we provide the regret guarantee for it. In particular, we show a regret of order $\mathcal{O}\left(d\sqrt{T}\log(\frac{TL^2}{\lambda\delta})\right)$ for SCLUCB, which has the same order as the lower bound proposed for LB in Dani et al. (2008); Rusmevichientong and Tsitsiklis (2010).

At each round $t$, given a regularized least-square (RLS) estimate of $\hat{\theta}_t$, SCLTS samples a perturbed parameter $\tilde{\theta}_t$ with an appropriate distributional property. Then, it searches for the action that maximizes the expected reward considering the parameter $\tilde{\theta}_t$ as the true parameter while respecting the safety constraint (2). If any such action exists, it is played under certain conditions; else, the algorithm resorts to playing a perturbed version of the baseline action that satisfies the safety constraint. In order to guarantee constraint satisfaction (a.k.a safety of actions), the algorithm builds a confidence region $\mathcal{E}_t$ that contains the unknown parameter $\theta_\star$ with high probability. Then, it constructs an *estimated safe* set $\mathcal{X}_t^s$ such that all actions $x_t \in \mathcal{X}_t^s$ satisfy the safety constraint for all $v \in \mathcal{E}_t$. The summary of the SCLTS presented in Algorithm 1, and a detailed explanation follows.

---

**Algorithm 1:** Stage-wise Conservative Linear Thompson Sampling (SCLTS)

---

1    **Input:** $\delta, T, \lambda, \rho_1$

2    Set $\delta' = \frac{\delta}{4T}$

3    **for** $t = 1, \ldots, T$ **do**

4        Sample $\eta_t \sim \mathcal{H}^{\text{TS}}$

5        Compute RLS-estimate $\hat{\theta}_t$ and $V_t$ according to (5)

6        Set $\tilde{\theta}_t = \hat{\theta}_t + \beta_t V_t^{-1/2} \eta_t$

7        Build the confidence region $\mathcal{E}_t(\delta')$ in (6)

8        Compute the estimated safe set $\mathcal{X}_t^s$ in (8)

9        **if** the following optimization is feasible: $x(\tilde{\theta}_t) = \text{argmax}_{x \in \mathcal{X}_t^s} \langle x, \tilde{\theta}_t \rangle$, **then**

10       Set $F = 1$, **else** $F = 0$

11       **if** $F = 1$ **and** $\lambda_{\min}(V_t) \geq \left( \frac{2L\beta_t}{\kappa_l + \alpha r_{b_l}} \right)^2$, **then**

12       Play $x_t = x(\tilde{\theta}_t)$

13       **else**

14       Play $x_t = (1 - \rho_1)x_{b_t} + \rho_1 \zeta_t$

15       Observe reward $y_t$ in (1)

16    **end for**

---

## 2.1   Algorithm description

Let $x_1, \ldots, x_t$ be the sequence of the actions and $r_1, \ldots, r_t$ be their corresponding rewards. For any $\lambda > 0$, we can obtain a regularized least-squares (RLS) estimate $\hat{\theta}_t$ of $\theta_\star$ as follows

$$\hat{\theta}_t = V_t^{-1} \sum_{s=1}^{t-1} y_s x_s, \text{ where } V_t = \lambda I + \sum_{s=1}^{t-1} x_s x_s^\top. \tag{5}$$

Algorithm 1 construct a confidence region

$$\mathcal{E}_t(\delta') = \mathcal{E}_t := \{ \theta \in \mathbb{R}^d : \|\theta - \hat{\theta}_t\|_{V_t} \leq \beta_t(\delta') \}, \tag{6}$$

where the ellipsoid radius $\beta_t$ is chosen according to the Proposition 2.1 in Abbasi-Yadkori et al. (2011) (restated below for completeness) in order to guarantee that $\theta_\star \in \mathcal{E}_t$ with high probability.

**Proposition 2.1.** *( Abbasi-Yadkori et al. (2011)) Let Assumptions 1, 2, and 3 hold. For a fixed $\delta \in (0, 1)$, and*

$$\beta_t(\delta) = R\sqrt{d \log \left( \frac{1 + \frac{tL^2}{\lambda}}{\delta} \right)} + \sqrt{\lambda}S \tag{7}$$

*with probability at least $1 - \delta$, it holds that $\theta_\star \in \mathcal{E}_t$.*

### 2.1.1   The estimated safe action set

Since $\theta_\star$ is unknown to the learner, she does not know whether an action $x \in \mathcal{X}$ is safe or not. Thus, she builds an estimated safe set such that each action $x_t \in \mathcal{X}_t^s$ satisfies the safety constraint for all $v \in \mathcal{E}_t$, i.e.,

$$\mathcal{X}_t^s := \{x \in \mathcal{X} : \langle x, v \rangle \geq (1 - \alpha)r_{b_t}, \forall v \in \mathcal{E}_t\} = \{x \in \mathcal{X} : \min_{v \in \mathcal{E}_t} \langle x, v \rangle \geq (1 - \alpha)r_{b_t}\} \tag{8}$$

$$= \{x \in \mathcal{X} : \langle x, \hat{\theta}_t \rangle - \beta_t(\delta')\|x\|_{V_t^{-1}} \geq (1 - \alpha)r_{b_t}\}. \tag{9}$$

Note that $\mathcal{X}_t^s$ is easy to compute since (9) involves a convex quadratic program. In order to guarantee safety, at each round $t$, the learner chooses her actions only from this estimated safe set in order to maximize the reward given the sampled parameter $\tilde{\theta}_t$, i.e.,

$$x(\tilde{\theta}_t) = \arg \max_{x \in \mathcal{X}_t^s} \langle x, \tilde{\theta}_t \rangle, \tag{10}$$

where $\tilde{\theta}_t = \hat{\theta}_t + \beta_t V_t^{-1/2}\eta_t$, and $\eta_t$ is a random IID sample from a distribution $\mathcal{H}^{\text{TS}}$ that satisfies certain distributional properties (see Abeille et al. (2017) or Defn. C.1 in Appendix C for more details). The challenge with $\mathcal{X}_t^s$ is that it contains actions which are safe with respect to all the parameters in $\mathcal{E}_t$, and not only $\theta_\star$. Hence, there may exist some rounds that $\mathcal{X}_t^s$ is empty. In order to face this problem, the algorithm proceed as follows. At round $t$, if the estimated action set $\mathcal{X}_t^s$ is not empty, SCLTS plays the safe action $x(\tilde{\theta}_t)$ in (10) only if the minimum eigenvalue of the Gram matrix $V_t$ is greater than $k_t^1 = \left(\frac{2L\beta_t}{\kappa_l + \alpha r_{b_t}}\right)^2$, i.e., $\lambda_{\min}(V_t) \geq k_t^1$, where $k_t^1$ is of order $\mathcal{O}(\log t)$. Otherwise, it plays the conservative action which is presented next. We show in Appendix C that $\lambda_{\min}(V_t) \geq k_t^1$ ensures that for the rounds that SCLTS plays the action $x(\tilde{\theta}_t)$ in (10), the optimal action $x_\star$ belongs to the estimated safe set $\mathcal{X}_t^s$, from which we can bound the regret of Term I in (12).

### 2.1.2 Conservative actions

In our setting, we assume that the learner is given a baseline policy that at each round $t$ suggests a baseline action $x_{b_t}$. We employ the idea proposed in Khezeli and Bitar (2019), which is merging the baseline actions with random exploration actions under stage-wise safety constraint. In particular, at each round $t$, SCLTS constructs a conservative action $x_t^{\text{cb}}$ as a convex combination of the baseline action $x_{b_t}$ and a random vector $\zeta_t$ as follows:

$$x_t^{\text{cb}} = (1 - \rho_1)x_{b_t} + \rho_1\zeta_t, \tag{11}$$

where $\zeta_t$ is assumed to be a sequence of independent, zero-mean and bounded random vectors. Moreover, we assume that $\|\zeta_t\|_2 = 1$ almost surely and $\sigma_\zeta^2 = \lambda_{\min}(\text{Cov}(\zeta_t)) > 0$. The parameters $\sigma_\zeta$ and $\rho_1$ control the exploration level of the conservative actions. In order to ensure that the conservative actions are safe, in Lemma 2.2, we establish an upper bound on $\rho_1$ such that for all $\rho_1 \in (0, \bar{\rho})$, the conservative action $x_t^{\text{cb}} = (1 - \rho_1)x_{b_t} + \rho_1\zeta_t$ is guaranteed to be safe.

**Lemma 2.2.** *At each round $t$, given the fraction $\alpha$, for any $\rho \in (0, \bar{\rho})$, where $\bar{\rho} = \frac{\alpha r_l}{S + r_h}$, the conservative action $x_t^{\text{cb}} = (1 - \rho)x_{b_t} + \rho\zeta_t$ is guaranteed to be safe almost surely.*

For the ease of notation, in the rest of this paper, we simply assume that $\rho_1 = \frac{r_l}{S + r_h}\alpha$.

At round $t$, SCLTS plays the conservative action $x_t^{\text{cb}}$ if the two conditions defined in Section 2.1.1 do not hold, i.e., either the estimated safe set $\mathcal{X}_t^s$ is empty or $\lambda_{\min}(V_t) < k_t^1$.

## 3 Regret Analysis

In this section, we provide a tight regret bound for SCLTS. In Proposition 3.1, we show that the regret of SCLTS can be decomposed into regret caused by choosing safe Thompson Sampling actions plus that of playing conservative actions. Then, we bound both terms separately. Let $N_{t-1}$ be the set of rounds $i < t$ at which SCLTS plays the action in (10). Similarly, $N_{t-1}^c = \{1, \ldots, t-1\} - N_{t-1}$ is the set of rounds $j < t$ at which SCLTS plays the conservative actions.

**Proposition 3.1.** *The regret of SCLTS can be decomposed into two terms as follows:*

$$R(T) \leq \underbrace{\sum_{t \in N_T} \left( \langle x_\star, \theta_\star \rangle - \langle x_t, \theta_\star \rangle \right)}_{\text{Term I}} + \underbrace{|N_T^c| \left( \kappa_h + \rho_1(r_h + S) \right)}_{\text{Term II}} \tag{12}$$

The idea of bounding Term I is inspired by Abeille et al. (2017): we wish to show that LTS has a constant probability of being "optimistic", in spite of the need to be conservative. In Theorem 3.2, we provide an upper bound on the regret of Term I which is of order $\mathcal{O}(d^{3/2}\log^{1/2} d\, T^{1/2}\log^{3/2} T)$.

**Theorem 3.2.** *Let $\lambda, L \geq 1$. On event $\{\theta_\star \in \mathcal{E}_t, \forall t \in [T]\}$, and under Assumption 4, we can bound Term I in (12) as:*

$$\text{Term I} \leq (\beta_T(\delta') + \gamma_T(\delta')(1 + \frac{4}{p}))\sqrt{2Td\log\left(1 + \frac{TL^2}{\lambda}\right)} + \frac{4\gamma_T(\delta')}{p}\sqrt{\frac{8TL^2}{\lambda}\log\frac{4}{\delta}}, \tag{13}$$

*where $\delta' = \frac{\delta}{6T}$, and $\gamma_t(\delta) = \beta_t(\delta')\left(1 + \frac{2}{C}\right)\sqrt{cd\log\left(\frac{c'd}{\delta}\right)}$*

We note that the regret of Term I has the same bound as that of Abeille et al. (2017) in spite of the additional safety constraints imposed on the problem. As the next step, in order to bound Term II in (12), we need to find an upper bound on the number of times $|N_T^c|$ that SCLTS plays the conservative actions up to time $T$. We prove an upper bound on $|N_T^c|$ in Theorem 3.3.

**Theorem 3.3.** *Let $\lambda, L \geq 1$. On event $\{\theta_\star \in \mathcal{E}_t, \forall t \in [T]\}$, and under Assumption 4, it holds that*

$$|N_T^c| \leq \left(\frac{2L\beta_T}{\rho_1 \sigma_\zeta (\kappa_l + \alpha r_l)}\right)^2 + \frac{2h_1^2}{\rho_1^4 \sigma_\zeta^4} \log(\frac{d}{\delta}) + \frac{2Lh_1\beta_T \sqrt{8\log(\frac{d}{\delta})}}{\rho_1^3 \sigma_\zeta^3 (\kappa_l + \alpha r_l)}, \tag{14}$$

*where $h_1 = 2\rho_1(1-\rho_1)L + 2\rho_1^2$ and $\rho_1 = (\frac{r_l}{S+r_h})\alpha$.*

**Remark 3.1.** *The upper bound on the number of times SCLTS plays the conservative actions up to time T provided in Theorem 3.3 has the order $\mathcal{O}\left(\frac{L^2 d \log(\frac{T}{\delta})\log(\frac{d}{\delta})}{\alpha^4 (r_l^2 \wedge r_l^4) \kappa_l (\sigma_\zeta^2 \wedge \sigma_\zeta^4)}\right)$.*

The first idea of the proof is based on the intuition that if a baseline action is played at round $\tau$, then the algorithm does not yet have a good estimate of the unknown parameter $\theta_\star$ and the safe actions played thus far have not yet expanded properly in all directions. Formally, this translates to small $\lambda_{\min}(V_\tau)$ and the upper bound $O(\log \tau) \geq \lambda_{\min}(V_\tau)$. The second key idea is to exploit the randomized nature of the conservative actions (cf. (11)) to lower bound $\lambda_{\min}(V_\tau)$ by the number of times $(N_\tau^c)$ that SCLTS plays the baseline actions up to that round (cf. Lemma D.1 in the Appendix). Putting these together leads to the advertised upper bound $O(\log T)$ on the total number of times $(N_T^c)$ the algorithm plays the baseline actions.

## 3.1 Additional Side Constraint with Bandit Feedback

We also consider the setting where the constraint depends on an unknown parameter that is different than the one in reward function. In particular, we assume the constraint of the form

$$\langle x_t, \mu_\star \rangle \geq (1-\alpha)q_{b_t}, \tag{15}$$

which needs to be satisfied by the action $x_t$ at every round $t$. In (15), $\mu_\star$ is a fixed, but unknown and the positive constants $q_{b_t} = \langle x_{b_t}, \mu_\star \rangle$ are known to the learner. In Section 4, we relax this assumption and we consider the case where the learner does not know the value of $q_{b_t}$. Let $\nu_{b_t} = \langle x_\star, \mu_\star \rangle - \langle x_{b_t}, \mu_\star \rangle$. Similar to Assumption 4, we assume there exist constants $0 \leq \nu_l \leq \nu_h$ and $0 < q_l \leq q_h$ such that $\nu_l \leq \nu_{b_t} \leq \nu_h$ and $q_l \leq q_{b_t} \leq q_h$.

We assume that with playing an action $x_t$, the learner observes the following bandit feedback:

$$w_t = \langle x_t, \mu_\star \rangle + \chi_t, \tag{16}$$

where $\chi_t$ is assumed to be a zero-mean $R$-sub-Gaussian noise. In order to handle this case, we show how SCLTS should be modified, and we propose a new algorithm called SCLTS-BF. The details on SCLTS-BF are presented in Appendix E. In the following, we only mention the difference of SCLTS-BF with SCLTS, and show an upper bound on its regret.

The main difference is that SCLTS-BF constructs two confidence regions $\mathcal{E}_t$ in (6) and $\mathcal{C}_t$ based on the bandit feedback such that $\theta_\star \in \mathcal{E}_t$ and $\mu_\star \in \mathcal{C}_t$ with high probability. Then, based on $\mathcal{C}_t$, it constructs the estimated safe decision set denoted $\mathcal{P}_t^s = \{x \in \mathcal{X} : \langle x, v \rangle \geq (1-\alpha)q_{b_t}, \forall v \in \mathcal{C}_t\}$. We note that SCLTS-BF only plays the actions from $\mathcal{P}_t^s$ that are safe with respect to all the parameters in $\mathcal{C}_t$.

We report the details on proving the regret bound for SCLTS-BF in Appendix E. We use the decomposition in Proposition 3.1, and we upper bound Term I similar to the Theorem 3.2. Then, we show an upper bound of order $\mathcal{O}\left(\frac{L^2 d \log(\frac{T}{\delta})\log(\frac{d}{\delta})}{\alpha^4 (q_l^2 \wedge q_l^4) \nu_l (\sigma_\zeta^2 \wedge \sigma_\zeta^4)}\right)$ over the number of times that SCLTS-BF plays the conservative actions.

## 4 Unknown Baseline Reward

Inspired by Kazerouni et al. (2017), which studies this problem in the presence of *safety constraints on the cumulative rewards*, we consider the case where the expected reward of the action chosen by baseline policy, i.e., $r_{b_t}$ is unknown to the learner. However, we assume that the learner knows the

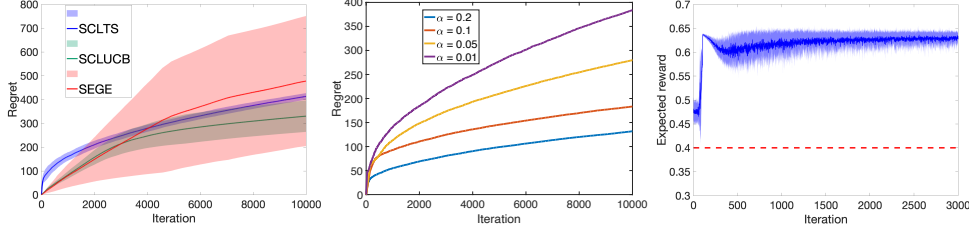

Figure 1: Left: comparison of the cumulative regret of SCLTS and SCLUCB versus SEGE algorithm in Khezeli and Bitar (2019). Middle: average regret (over 100 runs) of SCLTS algorithm for different values of $\alpha$. Right: expected reward under SCLTS algorithm in the first 3000 rounds for $\alpha = 0.2$.

value of $r_l$ in (4). We describe the required modifications on SCLTS to handle this case, and present a new algorithm called SCLTS2. Then, we prove the regret bound for SCLTS2, which has the same order as SCLTS.

Here, the learner does not know the value of $r_{b_t}$; however, she knows that the unknown parameter $\theta_\star$ falls in the confidence region $\mathcal{E}_t$ with high probability. Hence, we can upper bound the RHS of (2) with $\max_{v \in \mathcal{E}_t} \langle x_{b_t}, v \rangle \geq r_{b_t}$. Therefore, any action $x$ that satisfies

$$\min_{v \in \mathcal{E}_t} \langle x(\tilde{\theta}_t), v \rangle \geq (1 - \alpha) \max_{v \in \mathcal{E}_t} \langle x_{b_t}, v \rangle, \tag{17}$$

is safe with high probability. In order to ensure safety, SCLTS2 only plays the safe actions from the estimated safe actions set $\mathcal{Z}_t^s = \{ x \in \mathcal{X} : \min_{v \in \mathcal{E}_t} \langle x, v \rangle \geq (1 - \alpha) \max_{v \in \mathcal{E}_t} \langle x_{b_t}, v \rangle \}$. We report the details on SCLTS2 in Appendix F.

Next, we provide an upper bound on the regret of SCLTS2. To do so, we first use the decomposition in Proposition 3.1. The regret of Term I is similar to that of SCLTS (Theorem 3.2), and in Theorem 4.1, we prove an upper bound on the number of time SCLTS2 plays the conservative actions. Note that similar steps can be generalized to the setting of additional side constraints with bandit feedback.

**Theorem 4.1.** *Let $\lambda, L \geq 1$. On event $\{\theta_\star \in \mathcal{E}_t, \forall t \in [T]\}$, and under Assumption 4, we can upper bound the number of times SCLTS2 plays the conservative actions, i.e., $|N_T^c|$ as:*

$$|N_T^c| \leq \left( \frac{2L\beta_T(2-\alpha)}{\rho_3 \sigma_\zeta(\kappa_l + \alpha r_l)} \right)^2 + \frac{2h_3^2}{\rho_3^4 \sigma_\zeta^4} \log(\frac{d}{\delta}) + \frac{2Lh_3\beta_T(2-\alpha)}{\rho_3^3 \sigma_\zeta^3(\kappa_l + \alpha r_l)} \sqrt{8 \log(\frac{d}{\delta})}, \tag{18}$$

*where $h_3 = 2\rho_3(1 - \rho_3)L + 2\rho_3^2$ and $\rho_3 = (\frac{r_l}{S+1})\alpha$.*

**Remark 4.1.** *The regret of SCLTS2 has order of $\mathcal{O}\left( \frac{L^2 d \log(\frac{T}{\delta}) \log(\frac{d}{\delta})(2-\alpha)^2}{\alpha^4 (r_l^2 \wedge r_l^4) \kappa_l (\sigma_\zeta^2 \wedge \sigma_\zeta^4)} \right)$, which has the same rate as that of SCLTS. Therefore, the lack of information about the reward function only hurt the regret with a constant $(2 - \alpha)^2$.*

## 5 Numerical Results

In this section, we investigate the numerical performance of SCLTS and SCLUCB on synthetic data, and compare it with SEGE algorithm introduced by Khezeli and Bitar (2019). In all the implementations, we used the following parameters: $R = 0.1, S = 1, \lambda = 1, d = 2$. We consider the action set $\mathcal{X}$ to be a unit ball centered on the origin. The reward parameter $\theta_\star$ is drawn from $\mathcal{N}(0, I_4)$. We generate the sequence $\{\zeta_t\}_{t=1}^\infty$ to be IID random vectors that are uniformly distributed on the unit circle. The results are averaged over 100 realizations.

In Figure 1(left), we plot the cumulative regret of the SCLTS algorithm and SCLUCB and SEGE algorithm from Khezeli and Bitar (2019) for $\alpha = 0.2$ over 100 realizations. The shaded regions show standard deviation around the mean. In view of the discussion in Dani et al. (2008) regarding computational issues of LUCB algorithms with confidence regions specified with $\ell_2$-norms, we implement a modified version of Safe-LUCB which uses $\ell_1$-norms instead of $\ell_2$-norms. Figure 1(left) shows that SEGE algorithm suffers a high variance of the regret over different problem instances which shows the strong dependency of the performance of SEGE algorithm on the specific problem instance. However, the regret of SCLTS and SCLUCB algorithms do not vary significantly under

different problem instances, and has a low variance. Moreover, the regret of SEGE algorithm grows faster in the beginning steps, since it heavily relies on the baseline action in order to satisfy the safety constraint. However, the randomized nature of SCLTS leads to a natural exploration ability that is much faster in expanding the estimated safe set, and hence it plays the baseline actions less frequently than SEGE algorithm even in the initial exploration stages.

In Figure 1(middle), we plot the average regret of SCLTS for different values of $\alpha$ over a horizon $T = 10000$. Figure 1(middle) shows that, SCLTS has a better performance (i.e., smaller regret) for the larger value of $\alpha$, since for the small value of $\alpha$, SCLTS needs to be more conservative in order to satisfy the safety constraint, and hence it plays more baseline actions. Moreover, Figure 1(right) illustrates the expected reward of SCLTS algorithm in the first 3000 rounds. In this setting, we assume there exists one baseline action $x_b = [0.6, 0.5]$, which is available to the learner, $\theta_\star = [0.5, 0.4]$ and the safety fraction $\alpha = 0.2$. Thus, the safety threshold is $(1 - \alpha)\langle x_b, \theta_\star \rangle = 0.4$ (shown as a dashed red line), which SCLTS respects in all rounds. In particular, in initial rounds, SCTLS plays the conservative actions in order to respect the safety constraint, which as shown have an expected reward close to 0.475. Over time as the algorithm achieves a better estimate of the unknown parameter $\theta_\star$, it is able to play more optimistic actions and as such receives higher rewards.

# 6 Conclusion

In this paper, we study the stage-wise conservative linear stochastic bandit problem. Specifically, we consider safety constraints that requires the action chosen by the learner at each individual stage to have an expected reward higher than a predefined fraction of the reward of a given baseline policy. We propose extensions of Linear Thompson Sampling and Linear UCB in order to minimize the regret of the learner while respecting safety constraint with high probability and provide regret guarantees for them. We also consider the setting of constraints with bandit feedback, where the safety constraint has a different unknown parameter than that of the reward function, and we propose the SCLTS-BF algorithm to handle this case. Third, we study the case where the rewards of the baseline actions are unknown to the learner. Lastly, our numerical experiments compare the performance of our algorithm to SEGE of Khezeli and Bitar (2019) and showcase the value of the randomized nature of our exploration phase. In particular, we show that the randomized nature of SCLTS leads to a natural exploration ability that is faster in expanding the estimated safe set, and hence SCLTS plays the baseline actions less frequently as theoretically shown. For future work, natural extension of the problem setting to generalized linear bandits, and possibly with generalized linear constrains might be of interest.

# 7 Acknowledgment

This research is supported by NSF grant 1847096. C. Thrampoulidis was partially supported by the NSF under Grant Number 1934641.

# 8 Broader Impact

The main goal of this paper is to design and study novel "safe" learning algorithms for safety-critical systems with provable performance guarantees. An example arises in clinical trials where the effect of different therapies on patient's health is not known in advance. We select the baseline actions to be the therapies that have been historically chosen by medical practitioners, and the reward captures the effectiveness of the chosen therapy. The stage-wise conservative constraint modeled in this paper ensures that at each round the learner should choose a therapy which results in an expected reward if not better, must be close to the baseline policy. Another example arises in societal-scale infrastructure networks such as communication/power/transportation/data network infrastructure. We focus on the case where the reliability requirements of network operation at each round depends on the reward of the selected action and certain *baseline* actions are known to not violate system constraints and achieve certain levels of operational efficiency as they have been used widely in the past. In this case, the stage-wise conservative constraint modeled in this paper ensures that at each round, the reward of action employed by learning algorithm if not better, should be close to that of baseline policy in terms of network efficiency, and the reliability requirement for network operation must not be violated by the learner. Another example is in recommender systems that at each round, we wish to avoid recommendations that are extremely disliked by the users. Our proposed stage-wise conservative constrains ensures that at no round would the recommendation system cause severe dissatisfaction for the users (consider perhaps how a really bad personal movie recommendation from a streaming platform would severely affect your view of the said platform).

## Footnotes

[1]In Section 3.1, we show that our results also extend to constraints of the form $\langle x_t, \mu_\star \rangle \geq (1-\alpha)q_{b_t}$, where $\mu_\star$ is an additional unknown parameter. In this case, we assume the learner receives additional bandit feedback on the constraint after each round.

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
