[Supplementary Material]

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

# A  Proof of Proposition 3.1

From the definition of regret, we can write

$$R(T) = \sum_{t \in N_T} \left( \langle x_\star, \theta_\star \rangle - \langle x_t, \theta_\star \rangle \right) + \sum_{t \in N_T^c} \left( \langle x_\star, \theta_\star \rangle - \langle (1 - \rho_1) x_{b_t} - \rho_1 \zeta_t, \theta_\star \rangle \right)$$

$$= \sum_{t \in N_T} \left( \langle x_\star, \theta_\star \rangle - \langle x_t, \theta_\star \rangle \right) + \sum_{t \in N_T^c} \left( \langle x_\star, \theta_\star \rangle - \langle x_{b_t}, \theta_\star \rangle + \rho_1 \langle x_{b_t}, \theta_\star \rangle + \rho_1 \langle \zeta_t, \theta_\star \rangle \right)$$

$$\leq \sum_{t \in N_T} \left( \langle x_\star, \theta_\star \rangle - \langle x_t, \theta_\star \rangle \right) + |N_T^c| \left( \kappa_h + \rho_1 (r_h + S) \right). \tag{19}$$

# B  Proof of Lemma 2.2

In order to ensure that the conservative action $x_t = (1 - \rho) x_{b_t} + \rho \zeta_t$ is safe, we need to show that it satisfies (2). Hence, it suffices to show that

$$\langle (1 - \rho) x_{b_t} + \rho \zeta_t, \theta_\star \rangle \geq (1 - \alpha) r_{b_t}. \tag{20}$$

We can lower bound the LHS of (20) as follows:

$$\langle (1 - \rho) x_{b_t} + \rho \zeta_t, \theta_\star \rangle = r_{b_t} - \rho r_{b_t} + \rho \langle \zeta_t, \theta_\star \rangle \geq r_{b_t} - \rho r_{b_t} - \rho S.$$

Recall that $\|\zeta_t\|_2 = 1$ almost surely, and due to Assumption 2, we know that $\|\theta_\star\|_2 \leq S$. Hence, it suffices to show that

$$r_{b_t} - \rho r_{b_t} - \rho S \geq (1 - \alpha) r_{b_t},$$

or equivalently,

$$\rho r_{b_t} + \rho S \leq \alpha r_{b_t} \tag{21}$$

From (21) we can write

$$\rho \leq \frac{\alpha r_{b_t}}{S + r_{b_t}}. \tag{22}$$

Therefore, for any $\rho$ satisfying (22), the conservative action $x_t = (1 - \rho) x_{b_t} + \rho \zeta_t$ is guaranteed to be safe almost surely. Then, we lower bound the right hand side of (22) using Assumption 4, and we establish the following upper bound on $\rho$,

$$\rho \leq \frac{\alpha r_l}{S + r_h}. \tag{23}$$

Therefore, for any $\rho \in (0, \bar{\rho})$, where $\bar{\rho} = \frac{\alpha r_l}{S + r_h}$, the conservative actions are safe.

# C  Proof of Theorem 3.2

In this section, we provide an upper bound on the regret of Term I in (12). We first rewrite Term I as follows:

$$\sum_{t \in N_T} \left( \langle x_\star, \theta_\star \rangle - \langle x_t, \theta_\star \rangle \right) \tag{24}$$

Clearly, it would be beneficial to show that (24) is non-positive. However, as stated in Abeille et al. (2017) (in the case of linear TS applied to the standard stochastic linear bandit problem with no safety constraints), this cannot be the case in general. Instead, to bound regret in the unconstrained case, Abeille et al. (2017) argues that it suffices to show that (24) is non-positive with a constant probability. But what happens in the safety-constrained scenario? It turns out that once the above stated event happens with constant probability (in our case, in the presence of safety constraints), the rest of the argument by Abeille et al. (2017) remains unaltered. Therefore, our main contribution in the proof of Theorem 3.2 is to show that (24) is non-positive with a constant probability in spite of the limitations on actions imposed because of the safety constraints. To do so, let

$$\Theta_t^{\text{opt}} = \{ \theta \in \mathbb{R}^d : \langle x(\theta), \theta \rangle \geq \langle x_\star, \theta_\star \rangle \}, \tag{25}$$

be the so-called *set of optimistic parameters*, where $x(\tilde{\theta}_t) = \arg\max_{x \in \mathcal{X}_t^s} \langle x, \tilde{\theta}_t \rangle$ is the optimal safe action for the sampled parameter $\tilde{\theta}_t$ chosen from the estimated safe action set $\mathcal{X}_t^s$. LTS is considered optimistic at round $t$, if it samples the parameter $\tilde{\theta}_t$ from the set of optimistic parameters $\Theta_t^{\mathrm{opt}}$ and plays the action $x(\tilde{\theta}_t)$. In Lemma C.1, we show that SCLTS is optimistic with constant probability despite the safety constraints. Before that, let us restate the distributional properties put forth in Abeille et al. (2017) for the noise $\eta \sim \mathcal{H}^{\mathrm{TS}}$ that are required to ensure the right balance of exploration and exploitation.

**Definition C.1.** ( Definition 1. in Abeille et al. (2017)) $\mathcal{H}^{\mathrm{TS}}$ is a multivariate distribution on $\mathbb{R}^d$ absolutely continuous with respect to the Lebesgue measure which satisfies the following properties:

- (anti-concentration) there exists a strictly positive probability $p$ such that for any $u \in \mathbb{R}^d$ with $\|u\|_2 = 1$,

$$\mathbb{P}_{\eta \sim \mathcal{H}^{\mathrm{TS}}} \left( \langle u, \eta \rangle \geq 1 \right) \geq p. \tag{26}$$

- (concentration) there exists positive constants $c, c'$ such that $\forall \delta \in (0, 1)$

$$\mathbb{P}_{\eta \sim \mathcal{H}^{\mathrm{TS}}} \left( \|\eta\| \leq \sqrt{cd \log(\frac{c'd}{\delta})} \right) \geq 1 - \delta. \tag{27}$$

**Lemma C.1.** *Let* $\Theta_t^{opt} = \{ \theta \in \mathbb{R}^d : \langle x(\theta), \theta \rangle \geq \langle x_\star, \theta_\star \rangle \}$ *be the set of the optimistic parameters. For round* $t \in N_T$*, SCLTS samples the optimistic parameter* $\tilde{\theta}_t \in \Theta_t^{opt}$ *and plays the corresponding safe action* $x(\tilde{\theta}_t)$ *frequently enough, i.e.,*

$$\mathbb{P}(\tilde{\theta}_t \in \Theta_t^{opt}) \geq p. \tag{28}$$

*Proof.* We need to show that for rounds $t \in N_T$

$$\mathbb{P}\left( \langle x(\tilde{\theta}_t), \tilde{\theta}_t \rangle \geq \langle x_\star, \theta_\star \rangle \right) \geq p. \tag{29}$$

First, we show that for rounds $t \in N_T$, $x_\star$ falls in the estimated safe set, i.e., $x_\star \in \mathcal{X}_t^s$. To do so, we need to show that

$$\langle x_\star, \hat{\theta}_t \rangle - \beta_t \|x_\star\|_{V_t^{-1}} \geq (1 - \alpha) r_{b_t}, \tag{30}$$

using $\|\theta_\star - \hat{\theta}_t\|_{V_t} \leq \beta_t$, it suffices that

$$\langle x_\star, \theta_\star \rangle - 2\beta_t \|x_\star\|_{V_t^{-1}} \geq (1 - \alpha) r_{b_t}. \tag{31}$$

But we know that $\|x_\star\|_{V_t^{-1}} \leq \frac{\|x_\star\|_2}{\sqrt{\lambda_{\min}(V_t)}} \leq \frac{L}{\sqrt{\lambda_{\min}(V_t)}}$, where we also used Assumption 3 to bound $\|x_\star\|_2$. Hence, we can get

$$\langle x_\star, \theta_\star \rangle - 2\beta_t \|x_\star\|_{V_t^{-1}} \geq \langle x_\star, \theta_\star \rangle - \frac{2\beta_t L}{\sqrt{\lambda_{\min}(V_t)}}. \tag{32}$$

By substituting (32) in (31), it suffices to show that

$$\kappa_{b_t} + \alpha r_{b_t} \geq \frac{2\beta_t L}{\sqrt{\lambda_{\min}(V_t)}}, \tag{33}$$

or equivalently,

$$\lambda_{\min}(V_t) \geq \left( \frac{2L\beta_t}{\kappa_t + \alpha r_{b_t}} \right)^2. \tag{34}$$

To show (34), simply recall that $\lambda_{\min}(V_t) \geq k_t^1$, where $k_t^1 = \left( \frac{2L\beta_t}{\kappa_l + \alpha r_l} \right)^2$. Therefore, $x_\star \in \mathcal{X}_t^s$ for $t \in N_T$. Note that we are not interested in expanding the safe set in all possible directions. Instead, what aligns with the objective of minimizing regret, is expanding the safe set in the "correct" direction, that of $x_\star$. Therefore, $\lambda_{\min}(V_t) \geq \mathcal{O}(\log t)$ provides enough expansion of the safe set to bound the Term I in (12).

The rest of the proof is similar as in (Abeille et al., 2017, Lemma 3); we include in here for completeness.

For rounds $t \in N_T$, we know that

$$\langle x(\tilde{\theta}_t), \tilde{\theta}_t \rangle \geq \langle x_\star, \tilde{\theta}_t \rangle,$$

since $x(\tilde{\theta}_t) = \arg\max_{x \in \mathcal{X}_t^s} \langle x, \tilde{\theta}_t \rangle$ and we have already shown that $x_\star \in \mathcal{X}_t^s$. Therefore, it suffices to show that

$$\mathbb{P}\left( \langle x_\star, \tilde{\theta}_t \rangle \geq \langle x_\star, \theta_\star \rangle \right) \geq p. \tag{35}$$

From the definition of $\tilde{\theta}_t$, we can rewrite (35) as

$$\mathbb{P}\left( \langle x_\star, \hat{\theta}_t \rangle + \beta_t \langle x_\star, V_t^{-1/2} \eta_t \rangle \geq \langle x_\star, \theta_\star \rangle \right) \geq p,$$

or equivalently,

$$\mathbb{P}\left( \beta_t \langle x_\star, V_t^{-1/2} \eta_t \rangle \geq \langle x_\star, \theta_\star - \hat{\theta}_t \rangle \right) \geq p. \tag{36}$$

Then, we use Cauchy-Schwarz for the LHS of (36), and given the fact that $\|\theta_\star - \hat{\theta}_t\|_{V_t} \leq \beta_t$, we get

$$\mathbb{P}\left( \langle x_\star, V_t^{-1/2} \eta_t \rangle \geq \|x_\star\|_{V_t^{-1/2}} \right) \geq p,$$

or equivalently,

$$\mathbb{P}\left( \langle u_t, \eta_t \rangle \geq 1 \right) \geq p, \tag{37}$$

where $u_t = \frac{x_\star V_t^{-1/2}}{\|x_\star\|_{V_t^{-1/2}}}$. Therefore, $\|u_t\|_2 = 1$ by construction. At last, we know that (37) is true thanks to the anti-concentration distributional property of the parameter $\eta_t$ in Definition C.1. $\square$

As mentioned, after showing that SCLTS for rounds $t \in N_T$ samples from the set of optimistic parameters with a constant probability, the rest of the proof for bounding the regret of Term I is similar to that of Abeille et al. (2017). In particular, we conclude with the following bound

$$\text{Term I} := \sum_{t \in N_T} \left( \langle x_\star, \theta_\star \rangle - \langle x_t, \theta_\star \rangle \right)$$

$$(\beta_T(\delta') + \gamma_T(\delta')(1 + \frac{4}{p}))\sqrt{2|N_T|d\log\left(1 + \frac{|N_T|L^2}{\lambda}\right)} + \frac{4\gamma_T(\delta')}{p}\sqrt{\frac{8|N_T|L^2}{\lambda}\log\frac{4}{\delta}}, \tag{38}$$

where $\delta' = \frac{\delta}{6|N_T|}$, and,

$$\gamma_t(\delta) = \beta_t(\delta')\left(1 + \frac{2}{C}\right)\sqrt{cd\log\left(\frac{c'd}{\delta}\right)}, \tag{39}$$

and since $N_T \leq T$, the proof is completed.

# D   Proof of Theorem 3.3

In this section, we prove an upper bound of order $\mathcal{O}(\log T)$ on the number of times that SCLTS plays the conservative actions.

Let $\tau$ be any round that the algorithm plays the conservative action, i.e., at round $\tau$, either $F = 0$ or $\lambda_{\min}(V_\tau) < k_\tau^1 = \left(\frac{2L\beta_\tau}{\kappa_\tau + \alpha r_{b_\tau}}\right)^2$. By definition, if $F = 0$, we have

$$\nexists x \in \mathcal{X} : \langle x, \hat{\theta}_\tau \rangle - \beta_\tau \|x\|_{V_\tau^{-1}} \geq (1 - \alpha)r_{b_\tau}, \tag{40}$$

and since we know that $x_\star \in \mathcal{X}$, and $\theta_\star \in \mathcal{E}_t$ with high probability, we can write

$$\langle x_\star, \theta_\star \rangle - 2\beta_\tau \|x_\star\|_{V_\tau^{-1}} \leq \langle x_\star, \hat{\theta}_\tau \rangle - \beta_\tau \|x_\star\|_{V_\tau^{-1}} < (1 - \alpha)r_{b_\tau}. \tag{41}$$

From (41), we can get

$$\kappa_{b_\tau} + \alpha r_{b_\tau} < 2\beta_\tau \|x_\star\|_{V_\tau^{-1}} \leq \frac{2\beta_\tau L}{\sqrt{\lambda_{\min}(V_\tau)}}, \tag{42}$$

and hence the following upper bound on minimum eigenvalue of the Gram matrix:

$$\lambda_{\min}(V_\tau) < \left(\frac{2\beta_\tau L}{\kappa_{b_\tau} + \alpha r_{b_\tau}}\right)^2 \leq k_\tau^1. \tag{43}$$

Therefore, at any round $\tau$ that a conservative action is played, whether it is because $F = 0$, or because we have $\{\lambda_{\min}(V_\tau) < k_\tau\}$, we can always conclude that

$$\lambda_{\min}(V_\tau) < k_\tau^1. \tag{44}$$

The remainder of the proof builds on two auxiliary lemmas. First, in Lemma D.1, we show that the minimum eigenvalue of the Gram matrix $V_t$ is lower bounded with the number of times SCLTS plays the conservative actions.

**Lemma D.1.** *Under Assumptions 1, 2, and 3, it holds that*

$$\mathbb{P}(\lambda_{min}(V_t) \leq t) \leq d \exp\left(-\frac{(\rho_1^2 |N_t^c|\sigma_\zeta^2 - t)^2}{8|N_t^c|h_1^2}\right), \tag{45}$$

*where $h_1 = 2\rho_1(1 - \rho_1)L + 2\rho_1^2$ and $\rho_1 = (\frac{r_l}{S+r_h})\alpha$.*

Using (44) and applying Lemma D.1, it can be checked that with probability $1 - \delta$,

$$\left(\frac{2L\beta_\tau}{\kappa_l + \alpha r_l}\right)^2 > \rho_1^2 |N_\tau^c|\sigma_\zeta^2 - \sqrt{8|N_\tau^c|h_1^2 \log(\frac{d}{\delta})}. \tag{46}$$

This gives an explicit inequality that must be satisfied by $\tau$. Solving with respect to $\tau$ leads to the desired. In particular, we apply simple Lemma D.2 below.

**Lemma D.2.** *For any $a, b, c > 0$, if $ax - \sqrt{bx} < c$, then the following holds for $x \geq 0$*

$$0 \leq x < \frac{2ac + b + \sqrt{b^2 + 4abc}}{2a^2}. \tag{47}$$

Using Lemma D.2 results in the following upper bound on the $|N_\tau^c|$

$$|N_\tau^c| \leq \left(\frac{2L\beta_\tau}{\rho_1\sigma_\zeta(\kappa_l + \alpha r_l)}\right)^2 + \frac{2h_1^2}{\rho_1^4\sigma_\zeta^4}\log(\frac{d}{\delta}) + \frac{h_1 2L\beta_\tau}{(\kappa_l + \alpha r_l)\rho_1^3\sigma_\zeta^3}\sqrt{8\log(\frac{d}{\delta})}. \tag{48}$$

Therefore, we can upper bound $N_T^c$ with the following:

$$|N_T^c| \leq \left(\frac{2L\beta_T}{\rho_1\sigma_\zeta(\kappa_l + \alpha r_l)}\right)^2 + \frac{2h_1^2}{\rho_1^4\sigma_\zeta^4}\log(\frac{d}{\delta}) + \frac{2Lh_1\beta_T\sqrt{8\log(\frac{d}{\delta})}}{\rho_1^3\sigma^3(\kappa_l + \alpha r_l)}, \tag{49}$$

which has order $\mathcal{O}\left(\frac{L^2 d\log(\frac{T}{\delta})}{\alpha^2 r_l^2(\kappa_l + \alpha r_l)^2\sigma_\zeta^2} + \left(\frac{L^2}{\alpha^2 r_l^2\sigma_\zeta^4} + d^2\right)\log(\frac{d}{\delta})\right)$, as promised.

## D.1 Proof of Lemma D.1

Our objective is to establish a lower bound on $\lambda_{\min}(V_t)$ for all $t$. It holds that

$$
V_t = \lambda I + \sum_{s=1}^{t} x_s x_s^\top
$$

$$
\succeq \sum_{s \in N_t^c} \left( (1-\rho_1) x_{b_s} - \rho_1 \zeta_s \right) \left( (1-\rho_1) x_{b_s} - \rho_1 \zeta_s \right)^\top
$$

$$
= \sum_{s \in N_t^c} \left( (1-\rho_1)^2 x_{b_s} x_{b_s}^\top - \rho_1(1-\rho_1) x_{b_s} \zeta_s^\top - \rho_1(1-\rho_1) \zeta_s x_{b_s}^\top + \rho_1^2 \zeta_s \zeta_s^\top \right)
$$

$$
\succeq \sum_{s \in N_t^c} \left( -\rho_1(1-\rho_1) x_{b_s} \zeta_s^\top - \rho_1(1-\rho_1) \zeta_s x_{b_s}^\top + \rho_1^2 \zeta_s \zeta_s^\top \right)
$$

$$
= \sum_{s \in N_t^c} \left( \rho_1^2 \mathbb{E}[\zeta_s \zeta_s^\top] - \rho_1(1-\rho_1) x_{b_s} \zeta_s^\top - \rho_1(1-\rho_1) \zeta_s x_{b_s}^\top + \rho_1^2 \zeta_s \zeta_s^\top - \rho_1^2 \mathbb{E}[\zeta_s \zeta_s^\top] \right)
$$

$$
\succeq \rho_1^2 \sigma_\zeta^2 |N_t^c| I + \sum_{s \in N_t^c} U_s, \tag{50}
$$

where $U_s$ is defined as

$$
U_s = \left( -\rho_1(1-\rho_1) x_{b_s} \zeta_s^\top - \rho_1(1-\rho_1) \zeta_s x_{b_s}^\top + \rho_1^2 \zeta_s \zeta_s^\top - \rho_1^2 \mathbb{E}[\zeta_s \zeta_s^\top] \right). \tag{51}
$$

Then, using Weyl's inequality, it follows that

$$
\lambda_{\min}(V_t) \geq \rho_1^2 \sigma_\zeta^2 |N_t^c| - \lambda_{\max}\left( \sum_{s \in N_t^c} U_s \right).
$$

Next, we apply the matrix Azuma inequality (see Theorem D.3) to find an upper bound on $\lambda_{\max}(\sum_{s \in N_t^c} U_s)$. For this, we first need to show that the sequence of matrices $U_s$ satisfies the conditions of Theorem D.3. By definition of $U_s$ in (51), it follows that $\mathbb{E}[U_s | \mathcal{F}_{s-1}] = 0$, and $U_s^\top = U_s$. Also, we construct the sequence of deterministic matrices $A_s$ such that $U_s^2 \preceq A_s^2$ as follows. We know that for any matrix $B$, $B^2 \leq \|B\|_2^2 I$, where $\|B\|_2$ is the maximum singular value of $B$, i.e.,

$$
\sigma_{\max}(B) = \max_{\|u\|_1 = \|v\|_2 = 1} u^\top B v.
$$

Thus, we first show the following bound on the maximum singular value of the matrix $U_s$ defined in (51):

$$
\max_{\|u\|_1 = \|v\|_2 = 1} u^\top U_s v = -\rho_1(1-\rho_1)(u^\top x_{b_s})(v^\top \zeta_s)^\top - \rho_1(1-\rho_1)(u^\top \zeta_s)(v^\top x_{b_s})^\top +
$$

$$
\rho_1^2 (u^\top \zeta_s)(v^\top \zeta_s)^\top - \rho_1^2 \mathbb{E}\left[ (u^\top \zeta_s)(v^\top \zeta_s)^\top \right]
$$

$$
\leq \rho_1(1-\rho_1)\|x_{b_s}\|_2 \|\zeta_s\|_2 + \rho_1(1-\rho_1)\|\zeta_s\|_2 \|x_{b_s}\|_2 + \rho_1^2 \|\zeta_s\|_2^2 + \rho_1^2 \mathbb{E}\left[ \|\zeta_s\|_2^2 \right]
$$

$$
\leq 2\rho_1(1-\rho_1)L + 2\rho_1^2, \tag{52}
$$

where we have used Cauchy-Schwarz inequality and the last inequality comes from the fact that $\|\zeta_s\|_2 = 1$ almost surely, and $\|x_{b_s}\|_2 \leq L$ by Assumption 3. From the derivations above, and choosing $A_s = h_1 I$, with $h_1 = 2\rho_1(1-\rho_1)L + 2\rho_1^2$, it almost surely holds that $U_s^2 \preceq \sigma_{\max}(U_s)^2 I \preceq h_1^2 I = A_s^2$. Moreover, using triangular inequality, it holds that

$$
\| \sum_{s \in N_t^c} A_s^2 \| \leq \sum_{s \in N_t^c} \|A_s^2\| \leq |N_t^c| h_1^2.
$$

Now we apply the the matrix Azuma inequality, to conclude that for any $c \geq 0$,

$$
\mathbb{P}\left( \lambda_{\max}\left( \sum_{s \in N_t^c} U_s \right) \geq c \right) \leq d \exp\left( -\frac{c^2}{8|N_t^c| h_1^2} \right).
$$

Figure 2: Cumulative number of times that the baseline actions played by SCLTS up to time $t$, for $t = 1 \ldots, 1000$ over 100 realizations.

Therefore, it holds that with probability $1 - \delta$, $\lambda_{\max}(\sum_{s \in N_t^c} U_s) \leq \sqrt{8|N_t^c|h_1^2 \log(\frac{d}{\delta})}$, and hence with probability $1 - \delta$,

$$\lambda_{\min}(V_t) \geq \rho^2 |N_t^c| \sigma_\zeta^2 - \sqrt{8|N_t^c|h_1^2 \log(\frac{d}{\delta})},$$

or equivalently,

$$\mathbb{P}(\lambda_{\min}(V_t) \leq t) \leq d \exp\left( -\frac{(\rho_1^2 |N_t^c| \sigma_\zeta^2 - t)^2}{8|N_t^c|h_1^2} \right),$$

where $h_1 = 2\rho_1(1 - \rho_1)L + 2\rho_1^2$ and $\rho_1 = (\frac{r_l}{S + r_h})\alpha$. This completed the proof of lemma.

## D.2 Matrix Azuma Inequality

**Theorem D.3** (Matrix Azuma Inequality, Tropp (2012)). *Consider a sequence $\{Y_k\}$ of independent, random matrices adapted to the filtration $\{\mathcal{F}_k\}$. Each $\{Y_k\}$ is a self-adjoint matrix such that $\mathbb{E}[Y_k \,|\, \mathcal{F}_{k-1}] = 0$. Consider a fixed matrix $A_k$ such that $Y_k^2 \preceq A_k^2$ holds almost surely. Then, for $t \geq 0$, it holds that*

$$\mathbb{P}\left( \lambda_{max}\left( \sum_{k=1}^{s} Y_k \right) \geq t \right) \leq d \exp\left( -\frac{t^2}{8\| \sum_{k=1}^{s} A_k^2 \|} \right). \tag{53}$$

## D.3 Numerical analysis

In order to numerically verify our results in Theorem 3.3, we plot the cumulative number of time that baseline actions played bt SCLTS until time $t$ for $t = 1, \ldots, 1000$ over 100 realizations. The solid line in Figure 2 depicts average over 100 realizations and the shaded regions show standard deviation. The figure confirms the logarithmic trend predicted by theory.

# E  Upper Bounding the Regret of SCLTS-BF

In this section we provide the variation of our algorithm for the case of constraints with bandit feedback, which we refer to as SCLTS-BF in Algorithm 2. We then provide a regret bound for SCLTS-BF. The summary of SCLTS-BF is presented in Algorithm 2.

In this setting, we assume that at each round $t$, with playing an action $x_t$, the learner observes the reward $y_t = \langle x_t, \theta_\star \rangle + \xi_t$ and the following bandit feedback:

$$w_t = \langle x_t, \mu_\star \rangle + \chi_t, \tag{54}$$

where $\chi_t$ is assumed to be a zero-mean $R$-sub-Gaussian noise.

**Algorithm 2:** SCLTS-BF

---

17 **Input:** $\delta, T, \lambda, \rho$

18 Set $\delta' = \frac{\delta}{4T}$

19 **for** $t = 1, \ldots, T$ **do**

20      Sample $\eta_t \sim \mathcal{H}^{\mathrm{TS}}$

21      Compute RLS-estimate $\hat{\theta}_t$ and $V_t$ according to (5) and $\hat{\mu}_t$

22      Set $\tilde{\theta}_t = \hat{\theta}_t + \beta_t V_t^{-1/2} \eta_t$

23      Build the confidence region $\mathcal{E}_t(\delta')$ in (55) and $\mathcal{C}_t(\delta')$ in (56)

24      Compute the estimated safe set $\mathcal{P}_t^s = \{x \in \mathcal{X} : \langle x, v \rangle \geq (1 - \alpha)q_{b_t}, \forall v \in \mathcal{C}_t\}$

25      **if** the following optimization has a feasible solution: $x(\tilde{\theta}_t) = \mathrm{argmax}_{x \in \mathcal{P}_t^s} \langle x, \tilde{\theta}_t \rangle$, **then**

26      Set $F = 1$, **else** $F = 0$

27      **if** $F = 1$ **and** $\lambda_{\min}(V_t) \geq \left( \frac{2L\beta_t}{\nu_l + \alpha q_l} \right)^2$, **then**

28      Play $x_t = x(\tilde{\theta}_t)$

29      **else**

30      play $x_t = x_t^{\mathrm{cb}}$ defined in (59)

31      Observe reward $r_t$

32 **end for**

---

The main difference of SCLTS-BF with SCLTS is in the definition of the estimated safe action set. In particular, at each round $t$, SCLTS-BF constructs the following confidence regions:

$$\mathcal{E}_t(\delta') = \{\theta \in \mathbb{R} : \left\| \theta - \hat{\theta}_t \right\|_{V_t} \leq \beta_t(\delta')\}, \tag{55}$$

$$\mathcal{C}_t(\delta') = \{v \in \mathbb{R} : \|v - \hat{\mu}_t\|_{V_t} \leq \beta_t(\delta')\}, \tag{56}$$

where $\hat{\mu}_t = V_t^{-1} \sum_{s=1}^{t-1} w_s x_s$ is the RLS-estimate of $\mu_\star$. The radius in (55) and (56) is chosen according to Proposition 2.1 such that $\theta_\star \in \mathcal{E}_t$ and $\mu_\star \in \mathcal{C}_t$ with high probability. In order to ensure safety at each round $t$, SCLTS-BF constructs the following estimated safe action set

$$\mathcal{P}_t^s = \{x \in \mathcal{X} : \langle x, v \rangle \geq (1 - \alpha)q_{b_t}, \forall v \in \mathcal{C}_t\}. \tag{57}$$

The challenge with $\mathcal{P}_t^s$ is that it contains all the actions that are safe with respect to all the parameters in $\mathcal{C}_t$. Thus, there may exist some rounds that $\mathcal{P}_t^s$ is empty. To handle this case, SCLTS-BF proceed as follows. At each round $t$, given the sampled parameter $\tilde{\theta}_t$, if the estimated safe action set $\mathcal{P}_t^s$ defined in (57) is not empty, SCLTS-BF plays the safe action

$$x(\tilde{\theta}_t) = \arg \max_{x \in \mathcal{P}_t^s} \langle x, \tilde{\theta}_t \rangle \tag{58}$$

only if $\lambda_{\min}(V_t) \geq k_t^2$, where $k_t^2 = \left( \frac{2L\beta_t}{\nu_l + \alpha q_l} \right)^2$. Otherwise, it plays the following conservative action

$$x_t^{\mathrm{cb}} = (1 - \rho_2)x_{b_t} + \rho_2 \zeta_t, \tag{59}$$

where $\rho_2 = \alpha(\frac{q_l}{S + q_h})$ in order to ensure that the conservative actions are safe.

Next, we provide a regret guarantee for SCLTS-BF. First, we use the following decomposition of regret:

$$
\begin{aligned}
R(T) &= \sum_{t=1}^{T} \langle x_\star, \theta_\star \rangle - \langle x_t, \theta_\star \rangle \\
&= \underbrace{\sum_{t \in N_T} \left( \langle x_\star, \theta_\star \rangle - \langle x_t, \theta_\star \rangle \right)}_{\text{Term I}} + \underbrace{\sum_{t \in N_T^c} \left( \langle x_\star, \theta_\star \rangle - \langle (1 - \rho)x_{b_t} - \rho\zeta_t, \theta_\star \rangle \right)}_{\text{Term II}},
\end{aligned} \tag{60}
$$

where $N_t^c$ is the set of rounds $i < t$ that SCLTS-BF plays the conservative actions, and $N_t = \{1, \ldots, t\} - N_t^c$. In the following, we upper bound both Term I and Term II, separately.

**Bounding Term I.** Bounding Term I follows the same steps as that of Theorem 3.2. Here, we show that for SCLTS-BF, at rounds $t \in N_T$, the optimal action $x_\star$ belongs to the estimated safe safe, i.e., $x_\star \in \mathcal{P}_t^s$. Then, we conclude that regret of Term I similar to Theorem 3.2 has the order of $\mathcal{O}(d^{3/2} \log^{1/2} d \, T^{1/2} \log^{3/2} T)$.

At rounds $t \in N_T$, we know

$$\lambda_{\min}(V_t) \geq k_t^2 \geq \left( \frac{2L\beta_t}{\nu_{b_t} + \alpha q_{b_t}} \right)^2. \tag{61}$$

Then, in order to show that $x_\star \in \mathcal{X}_t^s$, we need to show

$$\langle x_\star, \hat{\mu}_t \rangle - \beta_t \|x_\star\|_{V_t^{-1}} \geq \langle x_\star, \mu_\star \rangle - 2\beta_t \|x_\star\|_{V_t^{-1}} \geq (1 - \alpha) q_{b_t}. \tag{62}$$

First inequality comes from the fact that $\|\mu_\star - \hat{\mu}_t\|_{V_t} \leq \beta_t$. Therefore, it suffices to show the second inequality holds. We use the fact that $\|x_\star\|_{V_t^{-1}} \leq \frac{\|x_\star\|_2}{\sqrt{\lambda_{\min}(V_t)}} \leq \frac{L}{\sqrt{\lambda_{\min}(V_t)}}$, where we use Assumption 3 to bound $\|x_\star\|_2$. Hence, we have

$$\langle x_\star, \mu_\star \rangle - 2\beta_t \|x_\star\|_{V_t^{-1}} \geq \langle x_\star, \mu_\star \rangle - \frac{2\beta_t L}{\sqrt{\lambda_{\min}(V_t)}}. \tag{63}$$

Then, it suffices to show that

$$\nu_{b_t} + \alpha q_{b_t} \geq \frac{2\beta_t L}{\sqrt{\lambda_{\min}(V_t)}}, \tag{64}$$

From (61), we know that (64) holds, and hence, $x_\star \in \mathcal{P}_t^s$. Therefore, we can use the result of Theorem 3.2, and obtain the desired regret bound.

**Bounding Term II.** First, we provide the formal statement of the theorem.

**Theorem E.1.** *Let $\lambda, L \geq 1$. On event $\left\{ \{\theta_\star \in \mathcal{E}_t, \forall t \in [T]\} \cap \{\mu_\star \in \mathcal{C}_t, \forall t \in [T]\} \right\}$, and Assumptions 4, we can upper bound the number of times SCLTS-BF plays the conservative actions, i.e., $|N_T^c|$ as:*

$$|N_T^c| \leq \left( \frac{2L\beta_T}{\rho_2 \sigma_\zeta (\alpha q_l + \nu_l)} \right)^2 + \frac{2h_2^2}{\rho_2^4 \sigma_\zeta^4} \log(\frac{d}{\delta}) + \frac{2Lh_2\beta_T \sqrt{8 \log(\frac{d}{\delta})}}{\rho_2^3 \sigma_\zeta^3 (\alpha q_l + \nu_l)} \tag{65}$$

*where $h_2 = 2\rho_2(1 - \rho_2)L + 2\rho_2^2$ and $\rho_2 = (\frac{q_l}{S+q_h})\alpha$.*

In order to prove Theorem E.1, we proceed as follows:

$$\sum_{t \in N_T^c} \left( \langle x_\star, \theta_\star \rangle - \langle (1 - \rho_2) x_{b_t} - \rho_2 \zeta_t, \theta_\star \rangle \right) = \sum_{t \in N_T^c} \langle x_\star, \theta_\star \rangle - \langle x_{b_t}, \theta_\star \rangle + \rho_2(\langle x_{b_t} + \zeta_t, \theta_\star \rangle)$$

$$\leq \sum_{t \in N_T^c} \nu_h + \rho_2(q_{b_t} + S) \leq |N_T^c|(\nu_h + \alpha q_l), \tag{66}$$

where $q_h \geq q_{b_t} \geq q_l > 0$ and $\nu_h \geq \nu_{b_t} \geq \nu_l$ for all $t$. Therefore, in order to bound Term II, it suffices to upper bound $|N_T^c|$ which is the number of rounds that SCLTS-BF plays the conservative actions up to round T. In order to do so, we proceed as follows:

Let $\tau$ be any round that the algorithm plays the conservative action.

If $F = 0$, i.e.,

$$\nexists x \in \mathcal{X} : \langle x, \hat{\mu}_\tau \rangle - \beta_\tau \|x\|_{V_\tau^{-1}} \geq (1 - \alpha) q_{b_\tau}, \tag{67}$$

and since we know that $x_\star \in \mathcal{X}$, and $\mu_\star \in \mathcal{C}_t$ with high probability, we can write

$$\langle x_\star, \mu_\star \rangle - 2\beta_\tau \|x_\star\|_{V_\tau^{-1}} \leq \langle x_\star, \hat{\mu}_\tau \rangle - \beta_\tau \|x_\star\|_{V_\tau^{-1}} < (1 - \alpha) q_{b_\tau}. \tag{68}$$

Using (68), we can get

$$\nu_{b_\tau} + \alpha q_{b_\tau} < 2\beta_\tau \|x_\star\|_{V_\tau^{-1}} \leq \frac{2\beta_\tau L}{\sqrt{\lambda_{\min}(V_\tau)}}, \tag{69}$$

and hence the following upper bound on minimum eigenvalue of the Gram matrix:

$$\lambda_{\min}(V_\tau) < \left(\frac{2\beta_\tau L}{\nu_{b_\tau} + \alpha q_{b_\tau}}\right)^2 \leq \left(\frac{2\beta_\tau L}{\nu_l + \alpha q_l}\right)^2 = k_\tau \tag{70}$$

Therefore, we show that in the cases where either the event $\{\nexists x \in \mathcal{X} : \langle x, \hat{\mu}_\tau \rangle - \beta_\tau \|x\|_{V_\tau^{-1}} \geq (1-\alpha)q_{b_\tau}\}$ or the event $\{\lambda_{\min}(V_\tau) < k_\tau^2\}$ happen, we can conclude that at round $\tau$

$$\lambda_{\min}(V_\tau) < k_\tau^2. \tag{71}$$

From Lemma D.1, we know that the minimum eigenvalue of the Gram matrix, i.e., $\lambda_{\min}(V_t)$ is lower bounded with the number of times that SCLTS-BF plays the conservative actions, i.e., $|N_T^c|$. Therefore, using (71), we can get

$$|N_T^c| \leq \left(\frac{2L\beta_T}{\rho_2 \sigma_\zeta (\alpha q_l + \nu_l)}\right)^2 + \frac{2h_2^2}{\rho_2^4 \sigma_\zeta^4} \log(\frac{d}{\delta}) + \frac{2Lh_2\beta_T\sqrt{2\log(\frac{d}{\delta})}}{\rho_2^3 \sigma_\zeta^3 (\alpha q_l + \nu_l)} \tag{72}$$

where $h_2 = 2\rho_2(1-\rho_2)L + 2\rho_2^2$ and $\rho_2 = \alpha(\frac{q_l}{S+q_h})$.

# F  Proof of Theorem 4.1

In this section, we first present the SCLTS2 algorithm, for the case where the learner does not know the reward of the actions suggested by baseline policy in advance, i.e., $r_{b_t}$. The summary of SCLTS2 is presented in Algorithm 3.

The algorithm relies on the fact that we can find an upper bound over the value of $r_{b_t}$, using the fact that $\theta_\star \in \mathcal{E}_t$, i.e.,:

$$\max_{v \in \mathcal{E}_t} \langle x_{b_t}, v \rangle \geq \langle x_{b_t}, \theta_\star \rangle = r_{b_t}. \tag{73}$$

Then, we can write the safety constraint as follows:

$$\min_{v \in \mathcal{E}_t} \langle x(\tilde{\theta}_t), v \rangle \geq (1-\alpha) \max_{v \in \mathcal{E}_t} \langle x_{b_t}, v \rangle. \tag{74}$$

It is easy to show that safety constraint (2) holds when (74) is true. Therefore, if we choose actions that satisfy (74), we can ensure that they are safe with respect to the safety constrain in (2).

Then we propose the estimated safe action set $\mathcal{Z}_t^s$ as:

$$\mathcal{Z}_t^s = \{x \in \mathcal{X} : \min_{v \in \mathcal{E}_t} \langle x, v \rangle \geq (1-\alpha) \max_{v \in \mathcal{E}_t} \langle x_{b_t}, v \rangle\}, \tag{75}$$

which contains actions that are safe with respect to all the parameter in $\mathcal{E}_t$. At each round $t$, SCLTS2 plays the safe action $x(\tilde{\theta}_t)$ from $\mathcal{Z}_t^s$ that maximizes the expected reward given the sampled parameter $\tilde{\theta}_t$, i.e.,

$$x(\tilde{\theta}_t) = \arg \max_{x \in \mathcal{Z}_t^s} \langle x, \tilde{\theta}_t \rangle \tag{76}$$

only if $\lambda_{\min}(V_t) \geq k_t^3$, where $k_t^3 = \left(\frac{2L\beta_t(2-\alpha)}{\kappa_l + \alpha r_l}\right)^2$. Otherwise it plays the conservative action $x_{b_t}^{cb}$ as:

$$x_t^{cb} = (1-\rho_3)x_{b_t} + \rho_3\zeta_t, \tag{77}$$

where $\rho_3 = \alpha(\frac{r_l}{S+1})$ such that the conservative action $x_t^{cb}$ is safe, where we use Assumption 3 for upper bounding the reward, i.e., $r_{b_t} \leq 1$.

In order to bound the regret of SCLTS2, we first use the decomposition defined in Proposition 3.1. The regret of Term I is similar to that of SCLTS (i.e., Theorem 3.2). Hence, it suffices to upper bound the number of time SCLTS2 plays the conservative actions, i.e., $|N_T^c|$.

**Algorithm 3:** SCLTS2

---

33 **Input:** $\delta, T, \lambda, \rho$
34 Set $\delta' = \frac{\delta}{4T}$
35 **for** $t = 1, \ldots, T$ **do**
36      Sample $\eta_t \sim \mathcal{H}^{\text{TS}}$
37      Compute RLS-estimate $\hat{\theta}_t$ and $V_t$ according to (5)
38      Set $\tilde{\theta}_t = \hat{\theta}_t + \beta_t V_t^{-1/2} \eta_t$
39      Build the confidence region $\mathcal{E}_t(\delta')$ in (6)
40      Compute the estimated safe set $\mathcal{Z}_t^s = \{x \in \mathcal{X} : \min_{v \in \mathcal{E}_t}\langle x, v\rangle \geq (1-\alpha)\max_{v \in \mathcal{E}_t}\langle x_{b_t}, v\rangle\}$
41      **if** the following optimization is feasible: $x(\tilde{\theta}_t) = \arg\max_{x \in \mathcal{Z}_t^s}\langle x, \tilde{\theta}_t\rangle$, **then**
42      Set $F = 1$, **else** $F = 0$
43      **if** $F = 1$ **and** $\lambda_{\min}(V_t) \geq \left(\frac{2L\beta_t(2-\alpha)}{\kappa_l + \alpha r_l}\right)^2$, **then**
44      Play $x_t = x(\tilde{\theta}_t)$
45      **else**
46      play $x_t = x_t^{\text{cb}}$ defined in (77)
47      Observe reward $y_t$
48 **end for**

---

In order to bound $|N_T^c|$, we proceed as follows. Let $\tau$ be the round that SCLTS2 plays a conservative action. If $F = 0$, i.e.,

$$\nexists x \in \mathcal{X} : \min_{v \in \mathcal{C}_\tau}\langle x, v\rangle \geq (1-\alpha)\max_{v \in \mathcal{C}_\tau}\langle x_{b_\tau}, v\rangle. \tag{78}$$

Using the fact that $x_\star \in \mathcal{X}$, we can write

$$\langle x_\star, \hat{\theta}_\tau\rangle - \beta_\tau\|x_\star\|_{V_\tau^{-1}} < (1-\alpha)\left(\langle x_{b_\tau}, \hat{\theta}_\tau\rangle + \beta_\tau\|x_{b_\tau}\|_{V_\tau^{-1}}\right). \tag{79}$$

Then, since $\|\theta_\star - \hat{\theta}_t\|_{V_t} \leq \beta_t$, we can upper bound the RHS and lower bound the LHS of (79), and get

$$\langle x_\star, \theta_\star\rangle - 2\beta_\tau\|x_\star\|_{V_\tau^{-1}} < (1-\alpha)\left(\langle x_{b_\tau}, \theta_\star\rangle + 2\beta_\tau\|x_{b_\tau}\|_{V_\tau^{-1}}\right), \tag{80}$$

or equivalently,

$$\kappa_{b_\tau} + \alpha r_{b_\tau} < 2\beta_\tau\|x_\star\|_{V_\tau^{-1}} + 2(1-\alpha)\beta_\tau\|x_{b_\tau}\|_{V_\tau^{-1}}. \tag{81}$$

Then we can use the fact that $\|x_\star\|_{V_\tau^{-1}} \leq \frac{L}{\sqrt{\lambda_{\min}(V_\tau)}}$ and $\|x_{b_\tau}\|_{V_\tau^{-1}} \leq \frac{L}{\sqrt{\lambda_{\min}(V_\tau)}}$, where we use Assumption 3 for upper bounding $\|x_\star\|_2$. Thus, we upper bound the RHS of (81) as follows:

$$\kappa_{b_\tau} + \alpha r_{b_\tau} < 2\beta_\tau\frac{L}{\sqrt{\lambda_{\min}(V_\tau)}} + 2(1-\alpha)\beta_\tau\frac{L}{\sqrt{\lambda_{\min}(V_\tau)}}, \tag{82}$$

and hence, we can get the following upper bound $\lambda_{\min}(V_\tau)$ as follows:

$$\lambda_{\min}(V_\tau) < \left(\frac{2L\beta_T(2-\alpha)}{\kappa_{b_\tau} + \alpha r_{b_\tau}}\right)^2 \leq \left(\frac{2L\beta_T(2-\alpha)}{\kappa_l + \alpha r_l}\right)^2 = k_\tau^3. \tag{83}$$

Therefore, we show that whether the event $F = 0$ happens or $\lambda_{\min}(V_t) < k_t^3$, we can achieve the upper bound provided in (83). Then, using the result of Lemma D.1, where we show that $\lambda_{\min}(V_t)$ is lower bounded with the number of times the algorithm plays the conservative actions, we obtain the following upper bound on the $|N_\tau^c|$

$$|N_\tau^c| \leq \left(\frac{2L\beta_\tau(2-\alpha)}{\rho_3\sigma_\zeta(\kappa_l + \alpha r_l)}\right)^2 + \frac{2h_3^2}{\rho_3^4\sigma_\zeta^4}\log(\frac{d}{\delta}) + \frac{2Lh_3\beta_\tau(2-\alpha)}{\rho_3^3\sigma_\zeta^3(\kappa_l + \alpha r_l)}\sqrt{2\log(\frac{d}{\delta})}, \tag{84}$$

where $h_3 = 2\rho_3(1-\rho_3)L + 2\rho_3^2$ and $\rho_3 = \alpha(\frac{r_l}{S+1})$.

# G Stage-wise Conservative Linear UCB (SCLUCB) Algorithm

In this section we propose a UCB-based safe stochastic linear bandit algorithm called Stage-wise Conservative Linear-UCB (SCLUCB), which is a safe counterpart of LUCB for the stage-wise conservative bandit setting. In particular, at each round $t$, given the RLS-estimate $\hat{\theta}_t$ of $\theta_\star$, SCLUCB constructs the confidence region $\mathcal{E}_t$ as follows:

$$\mathcal{E}_t(\delta) = \{\theta \in \mathbb{R}^d : \|\theta - \hat{\theta}_t\|_{V_t} \leq \beta_t(\delta)\}. \tag{85}$$

The radius $\beta_t(\delta)$ is chosen as in Proposition 2.1 such that $\theta_\star \in \mathcal{E}_t(\delta)$ with probability $1 - \delta$. Then, similar to SCLTS, it builds the estimated safe set $\mathcal{X}_t^s$ such that it includes actions that are safe with respect to all the parameter in $\mathcal{E}_t$, i.e.,

$$\mathcal{X}_t^s = \{x \in \mathcal{X} : \langle x, v \rangle \geq (1 - \alpha)r_{b_t}, \forall v \in \mathcal{E}_t\}. \tag{86}$$

Similar to SCLTS, the challenge with $\mathcal{X}_t^s$ is that there may exist some rounds that $\mathcal{X}_t^s$ is empty. In order to face this problem, SCLUCB proceed as follows. In order to guarantee safety, at each round $t$, if $\mathcal{X}_t^s$ is not empty, SCLUCB plays the action $\bar{x}_t$ as

$$(\bar{x}_t, \bar{\theta}_t) = \max_{x \in \mathcal{X}_t^s} \max_{v \in \mathcal{E}_t} \langle x, v \rangle \tag{87}$$

only if $\lambda_{\min}(V_t) \geq \left(\frac{2L\beta_t}{\kappa_l + \alpha r_{b_l}}\right)^2$, otherwise it plays the conservative action $x_t^{\text{cb}}$ defined in (11). The summary of SCLUCB is presented in Algorithm (4).

---

**Algorithm 4:** Stage-wise Conservative Linear UCB (SCLUCB)

---

49 **Input:** $\delta, T, \lambda, \rho$
50 **for** $t = 1, \ldots, T$ **do**
51      Compute RLS-estimate $\hat{\theta}_t$ and $V_t$ according to (5)
52      Build the confidence region $\mathcal{E}_t(\delta)$ in (85)
53      Compute the estimated safe set $\mathcal{X}_t^s$ in (86)
54      **if** the following optimization is feasible: $\bar{x}_t = \arg\max_{x \in \mathcal{X}_t^s} \max_{v \in \mathcal{E}_t} \langle x, v \rangle$, **then**
55      Set $F = 1$, **else** $F = 0$
56      **if** $F = 1$ **and** $\lambda_{\min}(V_t) \geq \left(\frac{2L\beta_t}{\kappa_l + \alpha r_{b_l}}\right)^2$, **then**
57      Play $x_t = \bar{x}_t$
58      **else**
59      play $x_t = x_t^{\text{cb}}$ defined in (11)
60      Observe reward $y_t$
61 **end for**

---

Next, we provide the regret guarantee for SCLUCB. Recall, $N_{t-1}$ be the set of rounds $i < t$ at which SCLUCB plays the action in (10). Similarly, $N_{t-1}^c = \{1, \ldots, t-1\} - N_{t-1}$ is the set of rounds $j < t$ at which SCLUCB plays the conservative actions.

**Proposition G.1.** *The regret of SCLUCB can be decomposed into two terms as follows:*

$$R(T) \leq \underbrace{\sum_{t \in N_T} (\langle x_\star, \theta_\star \rangle - \langle x_t, \theta_\star \rangle)}_{\text{Term I}} + \underbrace{|N_T^c| \left(\kappa_h + \rho_1(r_h + S)\right)}_{\text{Term II}} \tag{88}$$

In the following, we bound both terms, separately.

**Bounding Term I.** The first Term in (88) is the regret caused by playing the safe actions that maximize the reward given the true parameter is $\bar{\theta}_t$. The idea of bounding Term I is similar to Abbasi-Yadkori et al. (2011). We use the fact that for $t \in N_T$, $x_t = \bar{x}_t$, and start with the following decomposition of the instantaneous regret for $t \in N_T$ :

$$\langle x_\star, \theta_\star \rangle - \langle x_t, \theta_\star \rangle = \underbrace{\langle x_\star, \theta_\star \rangle - \langle \bar{x}_t, \bar{\theta}_t \rangle}_{\text{Term A}} + \underbrace{\langle \bar{x}_t, \bar{\theta}_t \rangle - \langle \bar{x}_t, \theta_\star \rangle}_{\text{Term B}} \tag{89}$$

**Bounding Term A.** Since for round $t \in N_t$, we require that $\lambda_{\min}(V_t) \geq k_t^1$, where $k_t^1 = \left(\frac{2L\beta_t}{\kappa_l + \alpha r_{b_l}}\right)^2$, we can conclude that $x_\star \in \mathcal{X}_t^s$. Therefore, due to (87), we have $\langle \bar{x}_t, \bar{\theta}_t \rangle \geq \langle x_\star, \theta_\star \rangle$, and hence Term A is not positive.

**Bounding Term B.** In order to bound Term B, we use the following chain of inequalities:

$$\begin{aligned}
\text{Term B} &:= \langle \bar{x}_t, \bar{\theta}_t \rangle - \langle \bar{x}_t, \theta_\star \rangle = \langle \bar{x}_t, \bar{\theta}_t \rangle - \langle \bar{x}_t, \hat{\theta}_t \rangle + \langle \bar{x}_t, \hat{\theta}_t \rangle - \langle \bar{x}_t, \theta_\star \rangle \\
&\leq \|\bar{x}_t\|_{V_t^{-1}} \|\bar{\theta}_t - \hat{\theta}_t\|_{V_t} + \|\bar{x}_t\|_{V_t^{-1}} \|\hat{\theta}_t - \theta_\star\|_{V_t} \\
&\leq 2\beta_t \|\bar{x}_t\|_{V_t^{-1}},
\end{aligned} \tag{90}$$

where the last inequality follows from Proposition 2.1. Recall, from Assumption 3, we have the following trivial bound:

$$\langle x_\star, \theta_\star \rangle - \langle \bar{x}_t, \theta_\star \rangle \leq 2. \tag{91}$$

Thus, we conclude the following

$$\text{Term B} \leq 2\min(\beta_t \|\bar{x}_t\|_{V_t^{-1}}, 1). \tag{92}$$

Next, we state a direct application of Lemma 11 in Abbasi-Yadkori et al. (2011).

**Lemma G.2.** *For $\lambda > 0$, and under Assumptions 1, 2, and 3, we have*

$$\sum_{t=1}^{T} \min(\|\bar{x}_t\|_{V_t^{-1}}^2, 1) \leq 2d\log\left(1 + \frac{TL^2}{\lambda d}\right) \tag{93}$$

Therefore, from Lemma G.2, we can conclude the following bound on regret of Term B:

$$\sum_{t \in N_T} 2\min(\beta_t \|\bar{x}_t\|_{V_t^{-1}}, 1) \leq 2\beta_T \sqrt{2d|N_T|\log(1 + \frac{|N_T|L^2}{\lambda d})}. \tag{94}$$

Next, in Theorem G.3, we provide an upper bound on the regret of Term I which is of order $\mathcal{O}\left(d\sqrt{T}\log(\frac{TL^2}{\lambda\delta})\right)$.

**Theorem G.3.** *On event $\{\theta_\star \in \mathcal{E}_t\}$ for a fixed $\delta \in (0, 1)$, with probability $1 - \delta$, it holds that:*

$$\sum_{t \in N_T} (\langle x_\star, \theta_\star \rangle - \langle x_t, \theta_\star \rangle) \leq 2\beta_T \sqrt{2dT\log(1 + \frac{TL^2}{\lambda d})} \tag{95}$$

**Bounding Term II.** In order to bound Term II in (88), we need to find an upper bound on the number of times that SCLUCB plays the conservative actions up to time $T$, i.e., $|N_T^c|$. We prove an upper bound on $|N_T^c|$ in Theorem G.4 which has the order of $\mathcal{O}\left(\frac{L^2 d\log(\frac{T}{\delta})\log(\frac{d}{\delta})}{\alpha^4(r_l^2 \wedge r_l^4)\kappa_l(\sigma_\zeta^2 \wedge \sigma_\zeta^4)}\right)$.

**Theorem G.4.** *Let $\lambda, L \geq 1$. On event $\{\theta_\star \in \mathcal{E}_t, \forall t \in [T]\}$, and under Assumption 4, we can upper bound the number of times SCLUCB plays the conservative actions, i.e., $|N_T^c|$ as:*

$$|N_T^c| \leq \left(\frac{2L\beta_T}{\rho_1 \sigma_\zeta(\kappa_l + \alpha r_l)}\right)^2 + \frac{2h_1^2}{\rho_1^4 \sigma_\zeta^4}\log(\frac{d}{\delta}) + \frac{2Lh_1\beta_T \sqrt{8\log(\frac{d}{\delta})}}{\rho_1^3 \sigma^3(\kappa_l + \alpha r_l)}, \tag{96}$$

*where $h_1 = 2\rho_1(1 - \rho_1)L + 2\rho_1^2$ and $\rho_1 = (\frac{r_l}{S+r_h})\alpha$.*

The proof is similar to that of Theorem 3.3, and we omit its proof here.

# H   Comparison with Safe-LUCB

In this section, we extend our results to an alternative safe bandit formulation proposed in Amani et al. (2019), where the algorithm Safe-LUCB was proposed. In order to do so, we first present the safety constraint in Amani et al. (2019), and then we show the required modification of SCLUCB to handle

this case, which we refer to as SCLUCB2. Then, we provide a problem-dependent regret bound for SCLUSB2, and we show that it matches the problem dependent regret bound of Safe-LUCB in Amani et al. (2019). We need to note that in Amani et al. (2019), they also provide a general regret bound of order $\tilde{\mathcal{O}}(T^{2/3})$ for Safe-LUCB which we do not discuss in this paper.

In Amani et al. (2019), it is assumed that the learner is given a convex and compact decision set $\mathcal{D}_0$ which contains the origin, and with playing the action $x_t$, she observes the reward of $y_t = x_t^\top \theta_\star + \eta_t$, where $\theta_\star$ is the fixed unknown parameter, and $\eta_t$ is $R$-sub-Gaussian noise. Moreover, The learning environment is subject to the linear safety constraint

$$x^\top B\theta_\star \leq C, \tag{97}$$

which needs to be satisfied at all rounds $t$ with high probability, and an action $x_t$ is called safe, if it satisfies (97). In (97), the matrix $B \in \mathbb{R}^{d \times d}$ and the positive constant $C$ are known to the learner. However, the learner does not receive any bandit feedback on the value $x^\top B\theta_\star$ and her information is restricted to those she receives from the reward.

Given the above constraint, the learner is restricted to choose actions from the safe set $\mathcal{D}_0^s$ as:

$$\mathcal{D}_0^s(\theta_\star) = \{x \in \mathcal{D}_0 : x^\top B\theta_\star \leq C\}. \tag{98}$$

Since $\theta_\star$ in unknown, the safe set $\mathcal{D}_0^s$ is unknown to the learner. Then, in Amani et al. (2019), they provide the problem-dependent regret bound (for the case where $\Delta := C - x^\top B\theta_\star > 0$) of order $\mathcal{O}(\sqrt{T}\log T)$. In the following, we present the required modification of SCLUSB to handle this safe bandit formulation, and propose the new algorithm called SCLUCB2 that we prove a problem dependnt regret bound of order $\mathbb{O}(\sqrt{T}\log T)$. We need to note that Amani et al. (2019) also provide a general regret bound of order $\tilde{\mathcal{O}}(T^{2/3})$ for the case where $\Delta = 0$; however, we do not discuss this case in this paper.

At each round $t$, given the RLS-estimate $\hat{\theta}_t$ of $\theta_\star$, SLUCB2 builds the confidence region $\mathcal{E}_t$ as:

$$\mathcal{E}_t = \{\theta \in \mathbb{R}^d : \|\theta - \hat{\theta}_t\|_{V_t} \leq \beta_t\}, \tag{99}$$

and the radius $\beta_t$ is chosen according to Proposition 2.1 such that $\theta_\star \in \mathcal{E}_t$ with high probability. The learner does not know the safe set $\mathcal{D}_0^s$; however, she knows that $\theta_\star \in \mathcal{E}_t$ with high probability. Hence, SLUCB2 constructs the estimated safe set $\mathcal{D}_t^s$ such that it contains actions that are safe with respect to all the parameter in $\mathcal{E}_t$, i.e.,

$$\begin{aligned}
\mathcal{D}_t^s &= \{x \in \mathcal{D}_0 : x^\top Bv \leq C, \forall v \in \mathcal{E}_t\} \\
&= \{x \in \mathcal{D}_0 : \max_{v \in \mathcal{E}_t} x^\top Bv \leq C\} \\
&= \{x \in \mathcal{D}_0 : x^\top B\hat{\theta}_t + \beta_t \|Bx\|_{V_t^{-1}} \leq C\}
\end{aligned} \tag{100}$$

Clearly, action $x = [0]^d$ (origin) is a safe action since $C > 0$, and also $[0]^d \in \mathcal{D}_0$. Thus, $[0]^d \in \mathcal{D}_t^s$. Since $x = [0]^d$ is a known safe action, we define the conservative action $x_0^c$ as:

$$x_0^c = (1 - \rho)[0]^d + \rho\zeta_t = \rho\zeta_t, \tag{101}$$

where $\zeta_t$ is a sequence of IID random vectors such that $\|\zeta_t\|_2 = 1$ almost surly, and $\sigma_\zeta = \lambda_{\min}(\text{Cov}(\zeta_t)) > 0$. We choose the constant $\rho$ according to the Lemma H.1 in order to ensure that the conservative action $x_0^c$ is safe.

**Lemma H.1.** *At each round $t$, for any $\rho \in (0, \bar{\rho})$, where*

$$\bar{\rho} = \frac{C}{\|B\|S}, \tag{102}$$

*the conservative action $x_0^c = \rho\zeta_t$ is guaranteed to be safe almost surly.*

We choose $\rho = \frac{C}{\|B\|S}$ for the rest of this section, and hence the conservative action is

$$x_0^c = \frac{C}{\|B\|S}\zeta_t. \tag{103}$$

Let $\Delta = C - x_\star^\top B\theta_\star$. We consider the case where $\Delta > 0$. At each $t$, in order to guarantee safety, SCLUCB2 only chooses its action from the estimated safe set $\mathcal{D}_t^s$. The challenge with $\mathcal{D}_t^s$ is that it includes actions that are safe with respect to all parameter in $\mathcal{E}_t$, and not only $\theta_\star$. Thus, there may exist some rounds that $\mathcal{D}_t^s$ is empty. At round $t$, if $\mathcal{D}_t^s$ is not empty, SCLUCB2 plays the safe action

$$\bar{x}_t = \arg \max_{x \in \mathcal{D}_t^s} \max_{v \in \mathcal{E}_t} \langle x, v \rangle \tag{104}$$

only if $\lambda_{\min}(V_t) \geq \left( \frac{2L\beta_t \|B\|}{\Delta} \right)^2$, otherwise it plays the conservative action $x_0^c$ in (103). The summary of SCLUCB2 is presented in Algorithm 5.

---

**Algorithm 5: SCLUCB2**

---

62 **Input:** $\delta, T, \lambda, \rho$
63 **for** $t = 1, \ldots, T$ **do**
64 $\quad$ Compute RLS-estimate $\hat{\theta}_t$ and $V_t$ according to (5)
65 $\quad$ Build the confidence region $\mathcal{E}_t(\delta)$ in (99)
66 $\quad$ Compute the estimated safe set $\mathcal{D}_t^s$ in (100)
67 $\quad$ **if** the following optimization is feasible: $\bar{x}_t = \arg \max_{x \in \mathcal{D}_t^s} \max_{v \in \mathcal{E}_t} \langle x, v \rangle$, **then**
68 $\quad$ Set $F = 1$, **else** $F = 0$
69 $\quad$ **if** $F = 1$ **and** $\lambda_{\min}(V_t) \geq \left( \frac{2L\beta_t \|B\|}{\Delta} \right)^2$, **then**
70 $\quad$ Play $x_t = \bar{x}_t$
71 $\quad$ **else**
72 $\quad$ play $x_t = x_0^c$ defined in (103)
73 $\quad$ Observe reward $y_t$
74 **end for**

---

In the following we provide the regret guarantee for SCLUCB2. Let $N_{t-1}$ be the set of rounds $i < t$ at which SCLUCB2 plays the action in (104). Similarly, $N_{t-1}^c = \{1, \ldots, t-1\} - N_{t-1}$ is the set of rounds $j < t$ at which SCLUCB2 plays the conservative action in (103).

First, we use the following decomposition of the regret, then we bound each term separately.

**Proposition H.2.** *The regret of SCLUCB2 can be decomposed to the following two terms:*

$$
\begin{aligned}
R(T) &= \sum_{t=1}^{T} \langle x_\star, \theta_\star \rangle - \langle x_t, \theta_\star \rangle \\
&= \sum_{t \in N_T} \left( \langle x_\star, \theta_\star \rangle - \langle x_t, \theta_\star \rangle \right) + \sum_{t \in N_T^c} \left( \langle x_\star, \theta_\star \rangle - \langle x_0^c, \theta_\star \rangle \right), \\
&\leq \underbrace{\sum_{t \in N_T} \left( \langle x_\star, \theta_\star \rangle - \langle x_t, \theta_\star \rangle \right)}_{\textit{Term I}} + \underbrace{2|N_t^c|}_{\textit{Term II}}.
\end{aligned}
\tag{105}
$$

**Bounding Term I.** In order to bound Term I, we proceed as follows. First, we show that at rounds $t \in N_T$, the optimal action $x_\star$ belongs to the estimated safe set $\mathcal{D}_t^s$, i.e., $x_\star \in \mathcal{D}_t^s$. To do so, we need to show that

$$x_\star^\top B\hat{\theta}_t + \beta_t \|Bx_\star\|_{V_t^{-1}} \leq C. \tag{106}$$

Since $\|\theta_\star - \hat{\theta}_t\|_{V_t} \leq \beta_t$, it suffices to show that:

$$x_\star^\top B\theta_\star + 2\beta_t \|Bx_\star\|_{V_t^{-1}} \leq C, \tag{107}$$

or equivalently

$$2\beta_t \|Bx_\star\|_{V_t^{-1}} \leq \Delta, \tag{108}$$

where $\Delta = C - x_\star^\top B\theta_\star$. It is easy to see (106) is true whenever (107) holds. Using Assumption 3, we can get $\|Bx_\star\|_{V_t^{-1}} \le \frac{\|B\|\|x_\star\|_2}{\sqrt{\lambda_{\min}(V_t)}} \le \frac{\|B\|L}{\sqrt{\lambda_{\min}(V_t)}}$. Hence, from (108), it suffices to show that

$$\frac{2\beta_t\|B\|L}{\sqrt{\lambda_{\min}(V_t)}} \le \Delta, \tag{109}$$

or equivalently

$$\lambda_{\min}(V_t) \ge \left(\frac{2\beta_t\|B\|L}{\Delta}\right)^2 \tag{110}$$

that we know it is true for $t \in N_T$. Therefore, on event $\{\theta_\star \in \mathcal{E}_t\}$, $x_\star \in \mathcal{D}_t^s$. We can bound the regret of Term I in (105) similar to Theorem G.3, and get the regret of order $\mathcal{O}\left(d\sqrt{T}\log(\frac{TL^2}{\lambda\delta})\right)$.

**Bounding Term II.** We need to upper bound the number of times that SCLUCB2 plays the conservative action $x_0^c$, i.e., $|N_T^c|$. We prove an upper bound on $|N_T^c|$ in Theorem H.3 which has the order of $\mathcal{O}\left(\frac{L^2 S^2 \|B\|^2 d \log(\frac{T}{\delta})\log(\frac{d}{\delta})}{\Delta^2(C\wedge C^2)(\sigma_\zeta^2 \wedge \sigma_\zeta^4)}\right)$.

**Theorem H.3.** *Let $\lambda, L \ge 1$. On event $\{\theta_\star \in \mathcal{E}_t, \forall t \in [T]\}$, we can upper bound the number of times SCLUCB2 plays the conservative actions, i.e., $|N_T^c|$ as:*

$$|N_T^c| \le \left(\frac{2LS\|B\|^2\beta_T}{C\Delta\sigma_\zeta}\right)^2 + \frac{32\log(\frac{d}{\delta})}{\sigma_\zeta^4} + \frac{8LS\|B\|^2\beta_T\sqrt{2\log(\frac{d}{\delta})}}{C\Delta\sigma_\zeta^3}. \tag{111}$$

*Proof.* Let $\tau$ be any round that the algorithm plays the conservative action, i.e., at round $\tau$, either $F = 0$ or $\lambda_{\min}(V_\tau) < \left(\frac{2L\|B\|\beta_\tau}{\Delta}\right)^2$.

By definition, if $F = 0$, we have

$$\nexists x \in \mathcal{X} : x^\top B\hat{\theta}_\tau + \beta_\tau\|Bx\|_{V_\tau^{-1}} \le C, \tag{112}$$

and since we know that $x_\star \in \mathcal{X}$, and $\theta_\star \in \mathcal{E}_t$ with high probability, we can write

$$x_\star^\top B\theta_\star + 2\beta_\tau\|Bx_\star\|_{V_\tau^{-1}} \ge x_\star^\top B\hat{\theta}_\tau + \beta_\tau\|Bx_\star\|_{V_\tau^{-1}} > C. \tag{113}$$

Then, using the LHS and RHS of (113), we can get

$$\frac{2L\|B\|\beta_\tau}{\sqrt{\lambda_{\min}(V_\tau)}} \ge 2\beta_\tau\|x_\star\|_{V_\tau^{-1}} \ge \Delta,$$

and hence the following upper bound on minimum eigenvalue of the Gram matrix:

$$\lambda_{\min}(V_\tau) < \left(\frac{2L\|B\|\beta_\tau}{\Delta}\right)^2.$$

Therefore, at any round $\tau$ that a conservative action is played, whether it is because $\{F = 0\}$ happens or beccause we have $\{\lambda_{\min}(V_\tau) < \left(\frac{2L\|B\|\beta_\tau}{\Delta}\right)^2\}$, we can always conclude that

$$\lambda_{\min}(V_\tau) < \left(\frac{2L\|B\|\beta_\tau}{\Delta}\right)^2 \tag{114}$$

The remaining of the proof builds on two auxiliary lemmas. First, in Lemma H.4, we show that the minimum eigenvalue of the Gram matrix $V_t$ is lower bounded with the number of times SCLUCB2 plays the conservative actions.

**Lemma H.4.** *On event $\{\theta_\star \in \mathcal{E}_t\}$, it holds that*

$$\mathbb{P}(\lambda_{min}(V_t) \le t) \le d\exp\left(-\frac{(\rho^2\sigma_\zeta^2|N_t^c| - t)^2}{32\rho^4|N_t^c|}\right), \tag{115}$$

*where $\rho = \frac{C}{\|B\|S}$.*

Using (114) and applying Lemma H.4, it can checked that with probability $1 - \delta$

$$\left(\frac{2L\|B\|\beta_\tau}{\Delta}\right)^2 > \rho^2 \sigma_\zeta^2 |N_\tau^c| - \sqrt{32\rho^4 . |N_\tau^c| \log(\frac{d}{\delta})},$$

Then using Lemma D.2, we can conclude the following upper bound

$$|N_\tau^c| \leq \left(\frac{2LS\|B\|^2\beta_\tau}{C\Delta\sigma_\zeta}\right)^2 + \frac{32\log(\frac{d}{\delta})}{\sigma_\zeta^4} + \frac{8LS\|B\|^2\beta_\tau\sqrt{2\log(\frac{d}{\delta})}}{C\Delta\sigma_\zeta^3}.$$

$\square$

## H.1 Proof of Lemma H.4

Our objective is to establish a lower bound on $\lambda_{\min}(V_t)$ for all $t$. It holds that

$$\begin{aligned}
V_t &= \lambda I + \sum_{s=1}^{t} x_s x_s^\top \\
&\succeq \sum_{s \in N_t^c} (\rho\zeta_s)(\rho\zeta_s)^\top \\
&= \sum_{s \in N_t^c} \left(\rho^2 \mathbb{E}[\zeta_s \zeta_s^\top] + \rho^2 \zeta_s \zeta_s^\top - \rho^2 \mathbb{E}[\zeta_s \zeta_s^\top]\right) \\
&\succeq \rho^2 \sigma_\zeta^2 |N_t^c| I + \sum_{s \in N_t^c} G_s,
\end{aligned} \tag{116}$$

where $G_s$ is defined as

$$G_s = \left(\rho^2 \zeta_s \zeta_s^\top - \rho^2 \mathbb{E}[\zeta_s \zeta_s^\top]\right). \tag{117}$$

Thus, using Weyl's inequality, it follows that

$$\lambda_{\min}(V_t) \geq \rho^2 \sigma_\zeta^2 |N_t^c| - \lambda_{\max}\left(\sum_{s \in N_t^c} G_s\right).$$

Next, we apply the matrix Azuma inequality (see Theorem D.3) to find an upper bound on $\lambda_{\max}(\sum_{s \in N_t^c} G_s)$. For this, we first need to show that the sequence of matrices $G_s$ satisfies the conditions of Theorem D.3. By definition of $G_s$ in (117), it follows that $\mathbb{E}[G_s|\mathcal{F}_{s-1}] = 0$, and $G_s^\top = G_s$. Also, we construct the sequence of deterministic matrices $A_s$ such that $G_s^2 \preceq A_s^2$ as follows. We know that for any matrix $K$, $K^2 \leq \|K\|_2^2 I$, where $\|K\|_2$ is the maximum singular value of $K$, i.e.,

$$\sigma_{\max}(K) = \max_{\|u\|_1 = \|v\|_2 = 1} u^\top K v.$$

Thus, we first show the following bound on the maximum singular value of the matrix $G_s$ defined in (117):

$$\begin{aligned}
\max_{\|u\|_1 = \|v\|_2 = 1} u^\top G_s v &= \rho^2 (u^\top \zeta_s)(v^\top \zeta_s)^\top - \rho^2 \mathbb{E}\left[(u^\top \zeta_s)(v^\top \zeta_s)^\top\right] \\
&\leq \rho^2 \|\zeta_s\|_2^2 + \rho^2 \mathbb{E}\left[\|\zeta_s\|_2^2\right] \\
&\leq 2\rho^2,
\end{aligned}$$

where we have used Cauchy-Schwarz inequality and the last inequality comes from the fact that $\|\zeta_s\|_2 = 1$ almost surely. From the derivations above, and choosing $A_s = 2\rho^2 I$, it almost surely holds that $G_s^2 \preceq \sigma_{\max}(G_s)^2 I \preceq 4\rho^4 I = A_s^2$. Moreover, using triangular inequality, it holds that

$$\|\sum_{s \in N_t^c} A_s^2\| \leq \sum_{s \in N_t^c} \|A_s^2\| \leq 4\rho^4 |N_t^c|.$$

Figure 3: Cumulative regret of SCLUCB2 versus Safe-LUCB in Amani et al. (2019) averaged over 100 realizations.

Now we can apply the matrix Azuma inequality, to conclude that for any $c \geq 0$,

$$\mathbb{P}\left(\lambda_{\max}(\sum_{s \in N_t^c} G_s) \geq c\right) \leq d \exp\left(-\frac{c^2}{32\rho^4|N_t^c|}\right).$$

Therefore, it holds that with probability $1 - \delta$, $\lambda_{\max}(\sum_{s \in N_t^c} G_s) \leq \sqrt{32\rho^4|N_t^c|\log(\frac{d}{\delta})}$, and hence with probability $1 - \delta$,

$$\lambda_{\min}(V_t) \geq \rho^2\sigma_\zeta^2|N_t^c| - \sqrt{32\rho^4|N_t^c|\log(\frac{d}{\delta})}, \tag{118}$$

or equivalently,

$$\mathbb{P}(\lambda_{\min}(V_t) \leq t) \leq d \exp\left(-\frac{(\rho^2\sigma_\zeta^2|N_t^c| - t)^2}{32\rho^4|N_t^c|}\right), \tag{119}$$

where $\rho = \frac{C}{\|B\|S}$. This completes the proof.

## H.2 Simulation Results

In order to verify our results on the regret bound of SCLUCB2, we plot the Figure 3 which plots the cumulative regret of the two algorithms averaged over 100 realizations. Therefore, the regret of SCLUCB2 matches the proposed problem-dependent upper bound in Amani et al. (2019).