[Reviews · NeurIPS 2020]

Review 1

Summary and Contributions: This paper presents linear bandit algorithms that ensures that some "safety" constraint (namely a lower bound on the instantaneous expected reward) is respected, by still trying to minimize the usual cumulative (pseudo-)regret. It offers the same regret bounds (O(sqrt(L)log(T)) as the state-of-the-art, while playing the non-optimal (safe) baseline actions only at most O(log(T)) times. This algorithm could be easily extended to other types of safety constraints, especially with unknown baseline rewards.

Strengths: * This work has good theoretical grounding and, to the best of my knowledge, is quite novel in the way it addresses an important conservativeness/safety constraint, expressed at the "stage"-level (i.e. for every time t, and not cumulated over time) * The approach is elegant in first estimating the safe action set and then in sampling an action in this safe set using a standard Thompson sampling strategy (when the safe set is not empty and an extra condition on the min eigenvalue of the Gram matrix is respected) or a perturbed -but safe - version of the baseline when this double condition is not met. * It can handle multiple types of safety constraints.

Weaknesses: * Relevant to a minority of the NeurIPS community * The algorithm needs a lot of assumptions (a lot of bounds are assumed to be known: S, L, R, r_l, r_h, \kappa_l, \kappa_h) : nothing is said about how to fix them or to which extent the algorithm is robust to mis-specified bounds. I'm sometimes doubting about the fact that these assumptions are valid in practice. * The same can be said about the hyper-parameters: there are numerous (\delta, \lambda, \rho, \alpha) and nothing is said about how to choose/tune them. * The experimental section is a bit disappointing. It is limited on synthetic data and on a simulation process that follows strictly the model assumptions. Only one setting/scenario is considered and authors compare results (cumulative regret ... but unfortunately not the number of times the safety constraint is violated) with only one competitor (SEGE [3]). It would have been interesting to compare SCLTS / SCLUCB with the method of [16], using the same setting as [16]. EDIT AFTER Rebuttal: @authors: thanks for your clear answers in the rebuttal. It would be great if you can include these answers (or elements of those) in the paper as well.

Correctness: Claims, theorems and proofs seem to be correct (to the best of my understanding and knowledge).

Clarity: The paper is well written: motivations are clearly exposed, ideas and methods are well structured and the paper is, globally, easy to read, except some inconsistencies in the notations (see Additional Feedback) and the constant need to refer to the Supplementary Material to understand important concepts).

Relation to Prior Work: To the best of my knowledge, the paper explains clearly the landscape of the current state-of-the-art and how the proposed approach differs from it, especially in terms of bounds and differences in the safety/conservativeness criteria.

Reproducibility: Yes

Additional Feedback: * Typo: psuedo-regret --> pseudo-regret * In footnote of page 2, $q_{b_t}$ is not defined. We have to wait for page 7 to know what it is. * S and L are actually linked through the constraint $<x,\theta_*> \leq 1$ . Please clarify. * Make clear in the assumptions that you need to know the constants (R,S,L, \kappa_h,\kappa_l, ...) in the algorithms, and not just to know that the quantities are bounded. *Please define $\delta'$ somewhere in between lines 170 and 174. * In Algo1 [line 6] and line 182, please make explicit the dependency of \beta on \delta' --> $\beta_t(\delta')$ * What is \rho used as input of Algorithm 1? It doesn't seem to be used.


Review 2

Summary and Contributions: The paper proposes variants of Thomson Sampling and UCB for the linear bandit setting, where the learner has to satisfy a safety constraint on the instantaneous reward in each round. Regret bounds (and constraint/safety guarantees) are provided and the rate matches the minimax rate achievable in linear bandits without constraints up to logarithmic factors. Several extensions of the setting are analyzed as well; and some empirical evaluation is presented.

Strengths: This setting of safe linear bandits has been studied before, and various results are known. The setting is important for many practical applications. The proposed algorithm and theory significantly improve upon previous known results in generality and the assumptions used are much more standard compared to previous work (in particular, a general compact action set is considered). The authors modify existing schemes (TS/UCB) in a simple way, which allows them to built on existing results for the un-constrained setting. Several interesting and relevant extensions are also worked out. The proofs of the regret bounds are clearly presented and follow known techniques, modula several modifications are necessary to account for the constraints.

Weaknesses: Assumption 4 is specific to the constrained setting. The authors should clarify to what extend these assumptions are necessary, and how the assumptions are restrictive. In particular, is it necessary that the base line policy achieves positive reward? Also, what is the benefit of introducing \kappa_l, if it is claimed that one can set \kappa_l=0 (line 4)? This seems to contradict Remark 3.1, where \kappa_l^{-1} appears in the sample complexity. However this is not evident from Theorem 3.3, and eq (49) also suggests otherwise. Further, the assumption 3 that X contains the unit ball could possibly hide some geometric complexity of the problem. For example, it does not allow a very "thin" action set, which would prevent the exploration method (11). Could the authors please comment on this? Simply re-scaling would affect other assumption such as <x, theta> < 1, so perhaps one should linearly transform the action set in this case (similar to John's position)?

Correctness: The technical results appear correct to me.

Clarity: The paper is well written and structured, for someone who is familiar with related literature, the ideas are easy to follow.

Relation to Prior Work: Yes, prior work is adequately discussed. One remark is that the authors describe related methods for Gaussian Processes as "non-linear bandit optimization with nonlinear constraints" (line 118). While it is true that kernelized bandits model non-linear functions in terms of the indexing domain, these methods heavily rely on linearity in the kernel space and analysis it typically not much different from the pure linear setting.

Reproducibility: Yes

Additional Feedback: Update: I have read the author's response and I will keep my score. One comment is that the term "LUCB" (which here is used for "Linear UCB" or "LinUCB") has been coined before with a different algorithm [1] Around line 59 it could be useful to mention why the knowledge of x_{b_t} is needed; at first sight it seems that (2) could be replaced by an generic lower bound. Algorithm 1, line 14: It would be useful to repeat the definition of x_t^{cb} [1] S. Kalyanakrishnan, A. Tewari, P. Auer, and P. Stone. PAC subset selection in stochastic multi-armed bandits. In International Conference on Machine Learning (ICML), 2012. Minor: line 55: X \in R -> X \subset R 68: leaner -> learner 72 & 74: psuedo -> pseudo


Review 3

Summary and Contributions: This paper studies the problem of stage-wise conservative linear bandits. Here, in addition to minimizing regret, the learner has to ensure that the regret at any time is no larger than a baseline threshold. The authors propose and analyze two algorithms.

Strengths: The paper is certainly relevant to the community. The algorithm improves upon the existing algorithms by playing the baseline arms O(log T) times as compared to O(sqrt{T}) times. The paper is well-written and clear to understand. The authors compare their work to [3] which originally formulated this problem and highlight the key differences in Sec 1.2. They also conduct experiments to compare the empirical performance of their algorithm to existing algorithms.

Weaknesses: There are several ways the paper can be improved. The fact that the baseline arm is played only log T times is an key property. It would be good to see this verified through experiments. I would also have liked to see a summary of the main points of the proof in the main paper. Finally, it would be good if authors can make the code public for reproducibility.

Correctness: Yes

Clarity: Yes.

Relation to Prior Work: Yes.

Reproducibility: Yes

Additional Feedback:

[Author Response · NeurIPS 2020]

We thank the reviewers for their constructive feedback and their valuable time.

**Reviewer 1:** First, we respond to the reviewer's **questions/suggestions regard-**
**ing the experimental results**. Regarding comparisons to the safe-LUCB of [16],
we present SCLUCB2 in App. H as a modified version of our main SCLUCB
algorithm tailored for the exact safe bandit setting studied in [16]. Importantly,
SCLUCB2 comes with a theoretical regret bound, which matches the proposed
problem-dependent upper bound in [16]. We now confirm this numerically in the
displayed figure, which plots the cumulative regret of the two algorithms averaged
over 100 realizations. We will include this new numerical study in the final version.
The reason that we only plot regret curves and not the number of times the safety constraint is violated, is because this
number is *zero* for almost all realizations. This is expected since all our algorithms guarantee the model's requirement
that the safety constraints are not violated for any time step, with high-probability $1 - \delta$. Second, **regarding the**
**parameters** $(R, S, L)$ **of Ass. 1-3**, assuming knowledge of them is standard in the literature of linear bandits (see
[5,11,10,13-15]). Their specific values are, of course, highly application-dependent, but the underlying hypothesis is
that they can be accurately determined based on domain-knowledge/physics, or, estimated from historic data. Even
if accurate approximations are not possible, rather loose bounds suffice to run the algorithms. Of course, the quality
of these bounds affects the performance, but, the accompanying regret-bounds quantify the effect. **Regarding the**
**parameters** $(r_l, r_h, \kappa_l, \kappa_h)$ **in Ass. 4** that are associated with the baseline policy, it can be reasonably assumed that they
can be estimated accurately from data. This is because we think of the baseline policy as "past strategy", implemented
before bandit-optimization, thus producing large amount of data (see also [1-3]). If no knowledge is available however,
$\kappa_h$ and $r_h$ can always be set to equal 1 (since for simplicity we assume that the mean rewards are in $[0, 1]$). Similarly, $\kappa_l$
can be set equal to zero. On a related note, we address the **question on tuning the hyper-parameters** $\delta, \lambda, \rho, \alpha$**.** The
tuning of $\delta, \lambda$ is standard and is same as in all linear-bandit algorithms [5,11,10,13-15]: $1 - \delta \in (0, 1)$ is the desired
confidence (e.g., 0.95) on the algorithm's realizations to satisfy the regret bounds (here, also the safety constraints); the
regularization parameter $\lambda$ can be set equal to one. The parameter $\rho$, controlling the exploration level of conservative
actions can take any value in the interval specified in Lemma 2.2. The parameter $\alpha \in (0, 1)$, controlling the conservatism
level of the learning process, is assumed known to the learner similar to [1,2,3]. We will clarify the above in the
revision. Finally, **the assumption** $\langle x, \theta_\star \rangle \leq 1$ is not essential and is rather only meant for simplicity. Specifically, the
assumptions $\|x\|_2 \leq L$ and $\|\theta_\star\|_2 \leq S$ suffice, as they guarantee the constant bound $LS$ for $\langle x, \theta_\star \rangle$; thus, nothing
fundamental changes in our analysis. For example, without this assumption, $\rho_3$ in Eq. (18) of Theorem 4.1. simply
changes to $\rho_3 = \frac{\alpha r_l}{S + LS}$. Contrary to our intention, this assumption appears to be confusing and will be removed in the
final version. Minor: The parameter $\rho$ in Algo. 1 appears in the definition of $x_t^{\text{cb}}$ in Eq. (11). We will clarify this.

**Reviewer 2:** First, please refer to lines 18-22 above on how the parameters ($r_l, r_h, \kappa_l, \kappa_h$) are chosen. We further
clarify the following. Regarding $\kappa_l$: Indeed, there is a typo in line 228 and the related factor in the sample complexity
should rather be $\kappa_l + \alpha r_l$ as specified explicitly in Thm. 3.3. What this bound suggests is that while setting $\kappa_l = 0$
is possible, a higher value is preferable (provided that it lower bounds $\kappa_{b_t}$ in (4)), since it results in smaller regret.
Regarding the requirement $r_l > 0$: Indeed, this is necessary for the algorithms to perform well and is critically
used in the proofs (e.g. Eq. (23)). That said, this is expected to be met in practice since the baseline policy is the
system's current strategy and should have been associated with at least a positive reward. Second, let us clarify the
**assumption that the action set contains the unit ball**, eqv. $L \geq 1$. This goes hand-in-hand with our assumption
$\|\zeta\|_2 = 1$ ( line 199), since together they guarantee that the convex combination $x_t^{\text{cb}}$ in (11) is feasible, i.e., satisfies
$\|x_t^{\text{cb}}\|_2 \leq L$. However, this requirement remains true as long as $\|\zeta\|_2 = \epsilon$ and $L \geq \epsilon$ for any $\epsilon > 0$. To see this, note
that $\|x_t^{\text{cb}}\|_2 \leq (1 - \rho)L + \rho\epsilon \leq L - \rho(L - \epsilon) \leq L$. In particular, $\epsilon > 0$ can be chosen small enough for "thin sets".
Changing $\|\zeta\|_2 = \epsilon$, we simply adjust $h_1 = 2\rho_1(1 - \rho_1)L\epsilon + 2\rho_1^2\epsilon^2$ and $\rho_1 = \frac{\alpha r_l}{S + r_h}$ in Thm. 3.3. Minor: Thank you
for the comment about LUCB. Also, we agree and will modify line 59 to clarify that the learner Knows $x_{b_t}$.

**Reviewer 3:** Thank you for the suggestion on **numerically verifying the number of**
**times that baseline is played.** The figure on the right plots the cumulative number

of baseline actions played by SCLTS until time $t$, for $t = 1, \ldots, 1000$. The solid line
depicts average over 100 realizations and the shaded regions show standard deviation.
The figure confirms the logarithmic trend predicted by theory. We will upload our code
as suggested. We finish with **a brief proof-sketch of Thm. 3.3**, which we will include
in the paper. The first idea is based on the intuition that if a baseline action is played at
round $t$, then the algorithm does not yet have a good estimate of the unknown parameter
$\theta_\star$ and the safe actions played thus far have not yet expanded properly in all directions.
Formally, this translates to small $\lambda_{\min}(V_t)$ and the *upper* bound $O(\log \tau) \geq \lambda_{\min}(V_\tau)$ (Eq. (43)). The second key idea
is to exploit the randomized nature of the conservative actions (cf. (11)) to *lower* bound $\lambda_{\min}(V_\tau)$ by the number $|N_\tau^c|$
of times that SCLTS plays the baseline actions up to that round (cf. Lemma D.1). Putting these together leads to the
advertised upper bound $O(\log T)$ on the total number $|N_T^c|$ of times the algorithm plays the baseline actions.

[Meta-Review · NeurIPS 2020]

The paper addresses issues of balancing exploration and exploitation when faced with a safety or minimum-utility requirement - a problem that has emerged recently within the online learning community. The reviewers all agree that the paper shows significant novelty and does a comprehensive job in terms of algorithm design and regret analysis. Some of the reviewers' concerns included the possibly restrictive nature of the assumptions made on the decision space and knowledge of problem-dependent structural constants, more explicit experiments to quantify the amount of exploration, and other technical clarifications, most of which were addressed by the author response satisfactorily. A broader, lingering concern that I have about the paper (and, in general, with this line of work) is that beyond the 'usual' application settings of recommendation systems and clinical trials, there is no attempt made to connect the technical formulation to a concrete and relatable problem, showing how the constraints actually arise in an organic manner. For instance, in clinical trials, the decision space (treatments) is likely finite, and low regret does not seem to be a logical objective compared to fast inference (best (safe) arm identification after an experimentation horizon). I would urge the author(s) to reflect upon this to situate the work better.